# A Closer Look at AUROC and AUPRC
# under Class Imbalance

**Matthew B. McDermott**
Harvard Medical School
matthew_mcdermott@hms.harvard.edu

**Haoran Zhang**
Massachusetts Institute of Technology
haoranz@mit.edu

**Lasse Hyldig Hansen**
Aarhus University
201908623@post.au.dk

**Giovanni Angelotti**
IRCCS Humanitas Research Hospital
giovanni.angelotti@humanitas.it

**Jack Gallifant**
Massachusetts Institute of Technology
jgally@mit.edu

## Abstract

In machine learning (ML), a widespread claim is that the area under the precision-recall curve (AUPRC) is a superior metric for model comparison to the area under the receiver operating characteristic (AUROC) for tasks with class imbalance. This paper refutes this notion on two fronts. First, we theoretically characterize the behavior of AUROC and AUPRC in the presence of model mistakes, establishing clearly that AUPRC is not generally superior in cases of class imbalance. We further show that AUPRC can be a *harmful metric* as it can unduly favor model improvements in subpopulations with more frequent positive labels, heightening algorithmic disparities. Next, we empirically support our theory using experiments on both semi-synthetic and real-world fairness datasets. Prompted by these insights, we reviewed over 1.5 million scientific papers to understand the origin of this invalid claim–finding it is often made without citation, misattributed to papers that do not argue this point, and aggressively overgeneralized from source arguments. Our findings represent a dual contribution: a significant technical advancement in understanding the relationship between AUROC and AUPRC and a stark warning about unchecked assumptions in the ML community.

## 1 Introduction

Machine learning (ML), especially in critical domains like healthcare, necessitates careful selection and application of evaluation metrics to guide appropriate model choices and understand performance nuances [150]. Model evaluation can happen in one of two settings: (1) a *methodological/model comparison* setting, which occurs outside of a specific deployment setting and in which target model usage workflows, optimal decision thresholds, or specific false-positive (FP) and false-negative (FN) costs are typically not known, or (2) an *application/deployment* setting, where reasonably specific estimates of model usage workflows and FP/FN costs can be made. In both settings, appropriate metric choice is critical, as inappropriate selection can hinder innovation when used for model comparison and lead to significant real-world costs (e.g., misdiagnosis in a medical setting) in deployment settings.

38th Conference on Neural Information Processing Systems (NeurIPS 2024).

This study focuses on two widely used metrics for binary classification tasks across both evaluation contexts: Area Under the Precision-Recall Curve (AUPRC) and Area Under the Receiver Operating Characteristic (AUROC). Central to this paper is the following key claim:

**Claim 1.** Let $f$ be a model which outputs continuous probabilistic predictions trained to solve a binary classification task for which the prevalence of negative labels is significantly higher than the prevalence of positive labels. For this problem, the AUPRC will yield a "better" or "more accurate" or "fairer" evaluation of $f$ than the AUROC.

Claim 1 is made widely in both the scientific literature [399, 71, 159, 124], in ML educational content [119, 141], and in popular press sources [80, 254]. It is so widespread that even basic search results for queries relating to AUROC and AUPRC[1] and large language model assistants like ChatGPT or Github Co-pilot will profess its veracity.[2] Throughout these sources, it has been justified on numerous, often imprecise grounds (see Section 5), but despite this extensive attention, *we show in this work that this claim is, in fact, wrong, and may be dangerous from a model fairness perspective*; further, many of its justifications are *invalid* or *misapplied* in common ML settings. More specifically, we show the following:

**1) AUROC and AUPRC only differ with respect to model-dependent parameters in that AUROC weighs all false positives equally, whereas AUPRC weighs false positives at a threshold $\tau$ with the inverse of the model's likelihood of outputting any scores greater than $\tau$ (Theorem 1).** This result shows that we can reason about the suitability of AUROC vs. AUPRC based on whether we care more about reducing false positives above low thresholds or high thresholds. In particular,

**2) AUROC favors model improvements uniformly over all positive samples, whereas AUPRC favors improvements for samples assigned higher scores over those assigned lower scores** (Theorem 2). This indicates that *the key factor differentiating the utility of AUROC or AUPRC as an evaluation metric is not class imbalance at all, but it is rather based on the target use case of the model in question.* See Figure 1 for a visual explanation. It also reveals that *AUPRC can amplify algorithmic biases.* In particular,

**3) AUPRC can unduly prioritize improvements to higher-prevalence subpopulations at the expense of lower-prevalence subpopulations**, raising serious fairness concerns in any multi-population use cases (Theorem 3).

In this work, we establish these three claims both theoretically and empirically via synthetic experiments and real-world validation on popular public fairness datasets. In addition, we demonstrate through an extensive, large-language model aided literature review of over 1.5 million scientific papers that Claim 1 has been used to motivate numerous improper uses of AUPRC relative to AUROC across high-stakes domains like healthcare and in several established venues, including AAAI, NeurIPS, ICML, ICLR, Cancer Cell, Nature Journals, PNAS, and more. Through this paper, we hope to shed light on the nuances of appropriate evaluation and provide key guidance to limit future misuse of evaluation metrics in the scientific and machine learning communities.

## 2 Theoretical Analyses

Please note that all notation used is defined in Appendix Section C.

### 2.1 Relationship between AUROC and AUPRC

In this section, we introduce Theorem 1, which is as follows:

*Theorem* 1. Let $\mathcal{X}, \mathcal{Y} = \{0, 1\}$ represent a paired feature and binary classification label space from which i.i.d. samples $(x, y) \in \mathcal{X} \times \mathcal{Y}$ are drawn via the joint distribution over the random variables $\mathsf{x}, \mathsf{y}$. Let $f : \mathcal{X} \to (0, 1)$ be a binary classification model outputting continuous probability scores over this space. Then,

$$\mathrm{AUROC}(f) = 1 - \mathbb{E}_{t \sim f(\mathsf{x})|\mathsf{y}=1} \left[ \mathrm{FPR}(f, t) \right]$$

$$\mathrm{AUPRC}(f) = 1 - P_{\mathsf{y}}(y = 0) \mathbb{E}_{t \sim f(\mathsf{x})|\mathsf{y}=1} \left[ \frac{\mathrm{FPR}(f, t)}{P_{\mathsf{x}}(f(x) > t)} \right]$$

---

[1]See `https://archive.is/qXPKu`, which shows results dominated by those making Claim 1

[2]See `https://chat.openai.com/share/f8f7fddb-1553-41a5-976d-789e2f3a90d6`

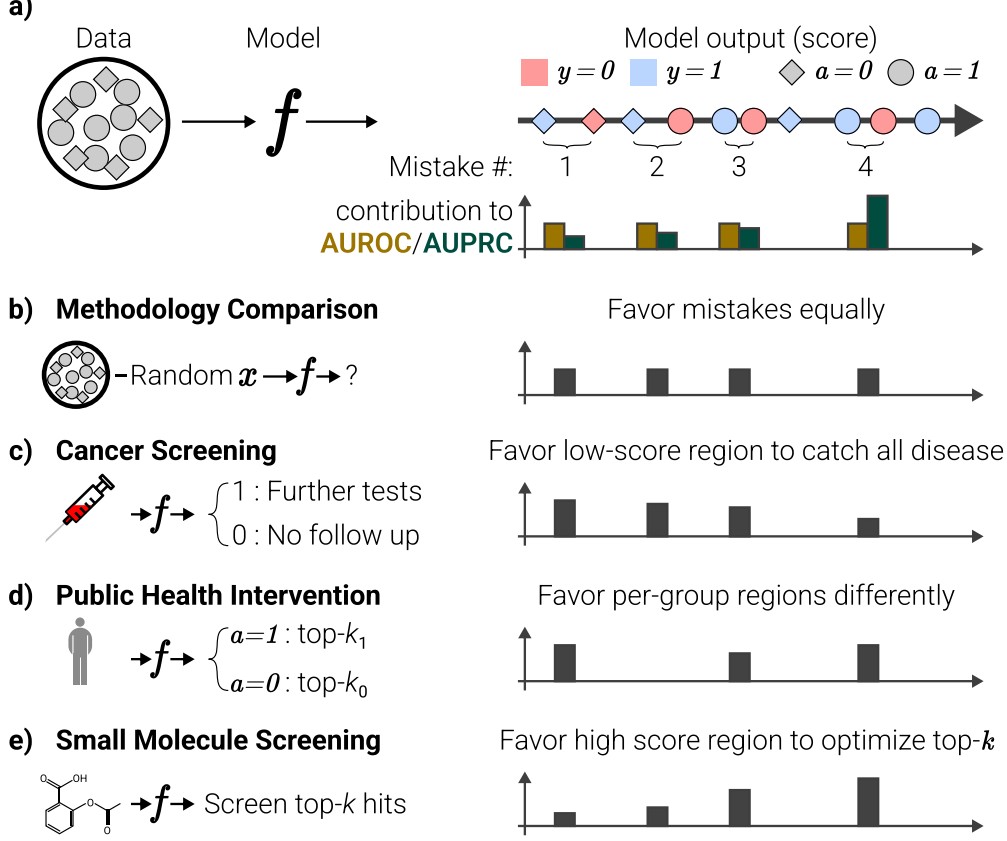

Figure 1: **a)** Consider a model $f$ yielding continuous output scores for a binary classification task applied to a dataset consisting of two distinct subpopulations, $\mathcal{A} \in \{0, 1\}$. If we order samples in ascending order of output score, each misordered pair of samples (e.g., mistake 1-4) represents an opportunity for model improvement. Theorem 2 shows that a model's AUROC will improve by the same amount no matter which mistake you fix, while the model's AUPRC will improve by an amount correlated with the score of the sample. **b)** When comparing models absent a specific deployment scenario, we have no reason to value improving one mistake over another, and model evaluation metrics should therefore improve equally regardless of which mistake is corrected. **c)** When false negatives have a high cost relative to false positives, evaluation metrics should favor mistakes that have *lower scores*, regardless of any class imbalance. **d)** When limited resources will be distributed among a population according to model score, *in a manner that requires certain subpopulations to all be offered commensurate possible benefit from the intervention for ethical reasons*, evaluation metrics should prioritize the importance of within-group, high-score mistakes such that the highest risk members of all subgroups receive interventions. **e)** When false positives are expensive relative to false negatives and there are no fairness concerns, evaluation metrics should favor model improvements in decreasing order with score.

We provide the proof in Appendix Section D. The two key intuitions are that integrating over the TPR is equivalent to taking the expectation over the induced distribution of positive sample scores, and that via Bayes rule, $\text{Prec}(f, \tau) = 1 - P_{\mathsf{y}}(y = 0)\frac{\text{FPR}(f,\tau)}{P_{\mathsf{x}}(f(x)>\tau)}$.

Despite its simplicity, Theorem 1 has far-reaching implications. Namely, it reveals that the only difference between AUROC and AUPRC with respect to model dependent parameters (i.e., omitting the dependence of AUPRC on the fixed prevalence of the dataset, which is not model varying) is that optimizing AUROC equates to minimizing the expected false positive rate over all positive samples in an unweighted manner (equivalently, in expectation over the distribution of positive sample scores) whereas optimizing AUPRC equates to minimizing the expected false positive rate over all positive samples weighted by the inverse of the model's "firing rate" ($P_{\mathsf{x}}(f(x) > \tau)$) at the given positive sample score. This preference can be crystallized when we examine how AUROC vs. AUPRC would prioritize correcting indivisible units of model improvements, termed "mistakes" which we will discuss next.

## 2.2 AUPRC prioritizes high-score mistakes, AUROC treats all mistakes equally

Understanding how a given evaluation metric prioritizes correcting various kinds of model mistakes or errors offers significant insight into when that metric should be used for optimization or model selection. To examine this topic for AUROC and AUPRC, consider the following definition of an "incorrectly ranked adjacent pair", which we will colloquially refer to as a "model mistake":

**Definition 2.1.** Let $f, \mathcal{X}, \mathcal{Y}, \mathsf{x}, \mathsf{y}$ be defined as in Theorem 1. Further, let us suppose we have sampled a static dataset from $\mathsf{x}, \mathsf{y}$ for evaluation which will be denoted $\boldsymbol{X}, \boldsymbol{y} = \{(x_1, y_1), \ldots, (x_N, y_N)\}$, for $x_i \in \mathcal{X}, y_i \in \{0, 1\}$, and $N \in \mathbb{N}$. We assume for convenience that $f$ is an injective map and all $x_i$ are distinct (i.e., $\forall (i, j) | i \neq j : x_i \neq x_j$ which, by injectivity of $f$, implies that $f(x_i) \neq f(x_j)$).

We say that $(x_i, x_j)$ are an *incorrectly ranked adjacent pair* and thus that the model makes a "*mistake*" at samples $(x_i, x_j)$ if:

1. $y_i = 1$ and $y_j = 0$
2. $f(x_i) < f(x_j)$
3. $\nexists x_k$ such that $f(x_i) < f(x_k) < f(x_j)$.

Essentially, Definition 2.1 states that a *mistake* occurs when a model assigns adjacent probability scores to a pair of samples with discordant labels, as shown in Figure 1. With this in mind, we can then introduce Theorem 2 which states that AUROC improves by a constant amount regardless of which mistake is corrected for a given model and dataset whereas AUPRC improves more when the mistake corrected occurs at a higher score than when it occurs at a lower score:

*Theorem* 2. Define $f, \mathcal{X}, \boldsymbol{X}, \boldsymbol{y}$ and $N$ as in Definition 2.1. Further, suppose without loss of generality that the dataset $\boldsymbol{X}$ is ordered such that $f(x_i) < f(x_{i+1})$ for all $i$. Then, let us define $M = \{i | (x_i, x_{i+1})$ is an *incorrectly ranked adjacent pair* for model $f\}$. Define $f_i'$ to be a model that is identical to $f$ except that the probabilities assigned to $x_i$ and $x_{i+1}$ are swapped:

$$f_i' : x \mapsto \begin{cases} f(x) & \text{if } x \notin \{x_i, x_{i+1}\} \\ f(x_{i+1}) & \text{if } x = x_i \\ f(x_i) & \text{if } x = x_{i+1}. \end{cases}$$

Then, $\text{AUROC}(f_i') = \text{AUROC}(f_j')$ for all $i, j \in M$, and $\text{AUPRC}(f_i') < \text{AUPRC}(f_j')$ for all $i, j \in M$ such that $i < j$.

The proof for Theorem 2 can be found in Appendix E. This proof simply stems from the fact that correcting a single mistake $(x_i, x_j)$ (as defined in Definition 2.1) always changes the false positive rate by the same amount, and only changes it at the threshold $f(x_i)$. This, combined with the formalization of AUROC and AUPRC in Theorem 1, establishes the proof. Note that this Theorem can be trivially extended to include a case where ties are possible simply by noting that "swapping" two samples $x_i$ and $x_j$ where $f(x_i) = f(x_j)$ in the manner of the theorem results in no change to either AUROC or AUPRC, and similarly by the same reasoning separating any tie in the appropriate direction will improve AUROC uniformly over samples and will improve AUPRC in a manner monotonic with model score.

## 2.3 AUPRC is explicitly discriminatory in favor of high-scoring subpopulations

The reliance on a model's firing rate revealed in Theorem 1 and the optimization behavior in Theorem 2 reveals significant issues with the fairness of AUPRC. In particular, in this section we introduce Theorem 3:

*Theorem* 3. Let $f, \mathcal{X}, \boldsymbol{X}, \boldsymbol{y}, N, M$, and $f'_j$ all be defined as in Theorem 2. Further, suppose that in this setting the domain $\mathcal{X}$ now contains an attribute defining two subgroups, $\mathcal{A} = \{0, 1\}$, such that for any sample $(x_i, y_i)$, $a_i$ denotes the subgroup to which that sample belongs. Let $f$ be perfectly calibrated for samples in subgroup $a = 0$, such that $P_{\mathsf{y}|\mathsf{a},\mathsf{x}}(y = 1 | a = 0, f(x) = t) = t$. Then,

$$\lim_{P_{\mathsf{y}|\mathsf{a}}(y=1|a=0) \to 0} P\left(a_i = a_{i+1} = 1 \middle| i = \underset{j \in M}{\arg\max}\left(\mathrm{AUPRC}(f'_j)\right)\right) = 1.$$

Essentially, Theorem 3 (proof provided in Appendix F) shows the following. Suppose we are training a model $f$ over a dataset with two subpopulations: Population $a = 0$ and $a = 1$. If the model $f$ is calibrated and the rate at which $y = 1$ for population $a = 0$ is sufficiently low relative to the rate at which $y = 1$ for population $a = 1$, then the mistake that, were it fixed, would maximally improve the AUPRC of $f$ will be a mistake purely in population $a = 1$. This demonstrates that AUPRC provably favors higher prevalence subpopulations (those with a higher base rate at which $y = 1$) under sufficiently severe prevalence imbalance between subpopulations.

Note that this property is, generally speaking, not desirable. *In particular, this property establishes that in settings where model fairness among a set of subpopulations in the data is important, AUPRC should not be used as an evaluation metric due to the risk that it will introduce biases in favor of the highest prevalence subpopulations.* We validate this result empirically over both synthetic and real-world data in Section 3, demonstrating that the import of Theorem 3 is not merely limited to an analytical curiosity but can have real-world impact on algorithmic disparities in practice. Furthermore, note that this theorem does not indicate that AUPRC will be superior to AUROC for *differentiating* a low prevalence (or low risk) subpopulation relative to a high-risk subpopulation, a property that is sometimes attributed to AUPRC in the literature. Rather, Theorem 3 shows that maximizing AUPRC will be more likely to optimize solely within the high-risk subgroup, rather than optimizing to differentiate across subgroups, as low-risk subgroup samples will predominantly occur in lower-score regions under severe class imbalance.

## 3 Experimental Validation

In this section, we establish via synthetic and real-world experiments that Theorem 3 is not merely an analytical effect but has real world consequences on the implications of optimizing or performing model selection via AUPRC.

### 3.1 Synthetic optimization experiments demonstrate AUPRC-induced disparities

In this section, we use a carefully constructed synthetic optimization procedure to demonstrate that, when all other factors are equal, optimizing by or performing model selection on the basis of AUPRC vs. AUROC risks excacerbating algorithmic disparities in the manner predicted by Theorem 3. For analyses under more realistic conditions with more standard models, see our real-world experiments in Section 3.2.

**Experimental Setup.** Let $y \in \{0, 1\}$ be the binary label, $s \in [0, 1]$ be the predicted score, and $a \in \{1, 2\}$ be the subpopulation. We fix $P_{\mathsf{y}|\mathsf{a}}(y = 1 | a = 1) = 0.05$ and $P_{\mathsf{y}|\mathsf{a}}(y = 1 | a = 2) = 0.01$. We sample a dataset for each group $\mathcal{D}_a = \{(s_1, y_1), ..., (s_{n_a}, y_{n_a})\}$, such that $\mathrm{AUROC}(\mathcal{D}_1) \approx \mathrm{AUROC}(\mathcal{D}_2) \approx \mathrm{AUROC}(\mathcal{D}_1 \cup \mathcal{D}_2) = 0.85$ (See Appendix G.1; A target AUROC of 0.65 was also profiled in Appendix Figure 5).

Our main experimental challenge is to determine how to simulate "optimizing" or "selecting" a model by AUROC or AUPRC. Simulating optimizing by these metrics allows us to explicitly assess how the use of either AUPRC or AUROC as an evaluation metric in model selection processes such as hyperparameter tuning or architecture search, can translate into model-induced inequities in dangerous ways. We explore two approaches here. First, we can simply correct the atomic mistake

that maximally improves AUROC or AUPRC in each optimization iteration. In our experiments, we use $n_1 = n_2 = 200$ and optimize for 50 steps for this experiment. This is the most straightforward optimization procedure to analyze, but it is unrealistic. In real optimization scenarios, larger model changes will be made at once, and a model will have an opportunity to *degrade* performance in some regions in order to improve it in others.

Next, we profile an optimization procedure that randomly permutes all the (sorted) model scores up to 3 positions (See Appendix G.3 for details). This has the effect of randomly adjusting all model scores, and can worsen model performance under some random permutations, but offers precisely the same "optimization capacity" to the low and high prevalence subgroups. To ensure the model is under some optimization constraint (and therefore does not always find the "perfect" permutation to maximize both metrics identically), we allow the model to sample only 15 possible permutations before choosing the best option. This means the system will be forced to navigate optimization trade-offs between which permutations improve the right regions of the score most effectively among its limited set. We use $n_1 = n_2 = 100$ for these experiments and optimize for 25 total steps.

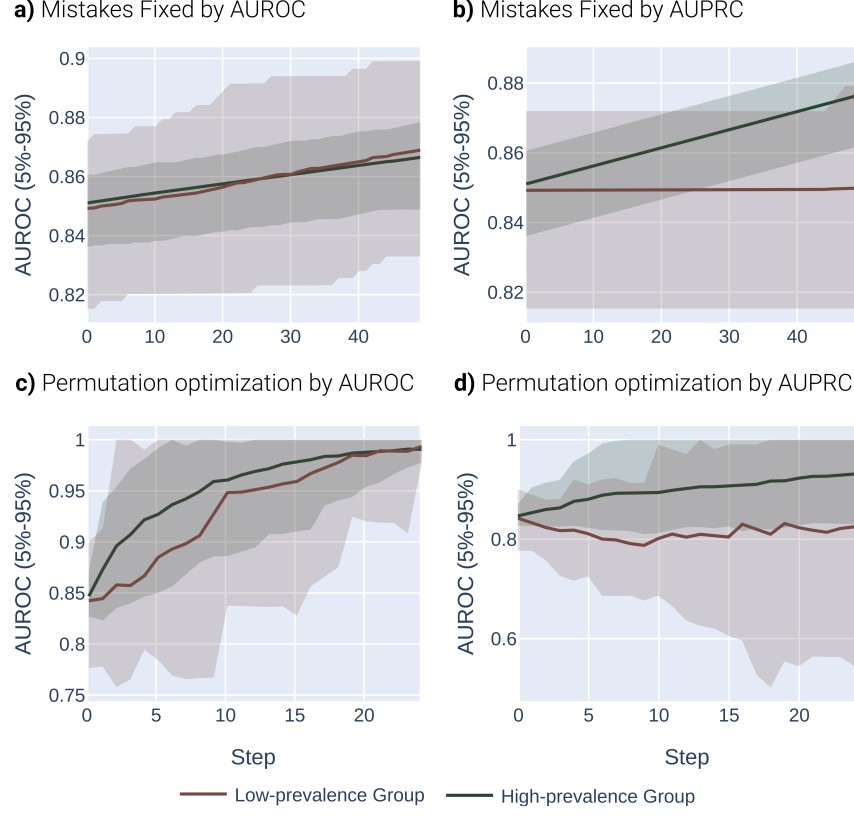

Figure 2: Synthetic experiment per-group AUROC, showing a confidence interval spanning the 5th to 95th percentile of results observed across all seeds, after successively either fixing individual mistakes, as defined in Definition 2.1, (**a**) and **b**)) or successively choosing the optimal score permutation (**c**) and **d**)) in order to optimize either AUROC (**a**) and **c**)) or AUPRC (**b**) and **d**)). It is clear across both forms of optimization that AUPRC definitively favors the higher prevalence subpopulation, whereas AUROC treats subgroups approximately equally. Similar patterns were observed when comparing per-group AUPRCs over the same experimental procedures, as shown in Appendix Figure 4.

Across both settings, we run these experiments across 20 randomly sampled datasets and show the mean and an empirical 90% confidence interval around the mean in Figure 2. We present a formal mathematical formulation of these perturbations, as well as profile a third random perturbation method, in Appendix G.3.

**Results.** Our results demonstrate the impact of the optimization metric on subpopulation disparity. In particular, in Figure 2, we observe a notable disparity introduced when optimizing under

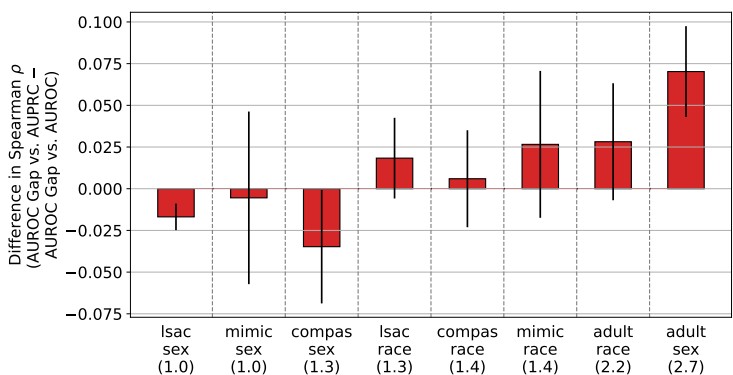

Figure 3: Difference in the Spearman's $\rho$ between the test-set signed AUROC gap versus the validation set overall AUPRC, and the AUROC gap versus the overall AUROC. Numbers in parentheses are the prevalence ratios between the two groups for the particular attribute, and datasets are sorted by this quantity. Error bars are 95% confidence intervals from 20 different random data splits.

the AUPRC metric regardless of the optimization procedure. This is evident in the performance metrics across the high and low prevalence subpopulations, which exhibit significant divergence as the optimization process favors the group with higher prevalence. In the more realistic, random-permutation optimization procedure (Figure 2d), this even results in a decrease in the AUROC for the low prevalence subgroup. In comparison, when optimizing for overall AUROC, the AUROCs of both groups increase together. Note that we show the effect of this optimization on the AUPRC metric, which shows very similar trends, in Appendix Figure 4. These results demonstrate explicitly that not only does optimizing for AUPRC differ greatly than for AUROC, as has been noted historically by researchers developing explicit AUPRC optimization schemes [409], but it in fact does so in an explicitly discriminatory way in realistic scenarios.

## 3.2 Real-world experimental validation

To demonstrate the generalizability of our findings to the real world, we evaluate fairness gaps induced by AUROC and AUPRC selection on four common datasets in the fairness literature [441, 99, 205].

**Datasets.** We use the following four tabular binary classification datasets: `adult` [17], `compas` [14], `lsac` [413], and `mimic` [178]. In each dataset, we consider both sex and race as sensitive attributes. To mimic the setting of our theorems, we balance each dataset by the sensitive attribute during both training and test, by randomly subsampling the majority group. Further details about each dataset, as well as preprocessing steps, can be found in Appendix H.

**Experimental setup.** We train XGBoost models [65] on each dataset. For each task, we iterate over a grid of per-group weights in order to create a diverse set of models that favor different groups. For each setting of task and per-group weight, we conduct a random hyperparameter search [37] with 50 runs. Though more complex hyperparameter search methods such as BOHB [100] or TPE [36] might lead to better performance, random searches are far more popular and practical, and have been used in popular benchmarking libraries [131, 337].

We evaluate the validation set overall AUROC and AUPRC. We also evaluate the test set AUROC gap and AUPRC gap between groups, where gaps are defined as the value of the metric for the higher prevalence group minus the value for the lower prevalence group. Based on our theorems, our hypothesis is that overall AUPRC should be more positively correlated with the signed AUROC gap than overall AUROC, indicating that it better favors the higher prevalence group, especially when the prevalence ratio between groups is high. To test this hypothesis, we evaluate the Spearman correlation coefficient between these quantities. We repeat this experiment 20 times, with different random data splits, to obtain a 95% confidence interval.

**Results.** In Figure 3, we plot the difference in the Spearman correlation coefficient of the AUROC gap versus the overall AUPRC, and AUROC gap versus overall AUROC. We observe mixed results in datasets with low prevalence ratio. In dataset with higher prevalence ratio, we find that overall AUPRC

is more positively correlated with the AUROC gap than overall AUROC, indicating that AUPRC more aggressively favors the higher prevalence group. We emphasize that the prevalence ratios observed in these real-world datasets is much lower than the ratio of 5 used in our synthetic experiments, which may account for the mild effect observed. To see raw results from these experiments, see Appendix Figure 7. Similar results for neural network classifiers can be found in Appendix Section H.3.

Next, in Appendix Figure 8, we plot the difference in the Spearman's $\rho$ from Figure 3, versus the prevalence gap. We find that there is a statistically significant correlation between the two (Spearman's $\rho = 0.905$, p = 0.002). Thus, while our power to detect a prevalence mediated AUPRC bias amplification effect is limited due to the limited prevalence disparities in these datasets, we nonetheless observe a strong positive correlation between the extent of the prevalence mismatch between the low and high prevalence group and the amount that AUPRC favors the high prevalence group over AUROC. *In other words, our results show that across these fairness datasets and attributes, as the prevalence disparity grows more extreme, we observe a statistically significant corresponding increase in the extent to which AUPRC introduces algorithmic bias, exactly in accordance with what Theorem 3 suggests.*

# 4 When *Should* One Use AUPRC vs. AUROC?

In Sections 2 and 3, we have shown that AUPRC is not universally superior in cases of class imbalance (and that instead, it merely preferentially optimizes high-score regions over low-score regions) and that it also poses serious risks to the model fairness in settings where subgroup prevalences differ. In light of this, how should we revise Claim 1 to reflect when we *actually* should use AUPRC instead of AUROC or vice versa? Below, we explore this question and provide practical guidance on metric selection for binary classification, building on our theoretical results and highlight specific contexts in which one may be favorable (Figure 1). Note that while we provide guidance below on situations in which AUROC vs. AUPRC is more or less favorable, this is not to suggest that authors should not report both metrics, or even larger sets of metrics or more nuanced analyses such as ROC or PR curves; rather this section is intended to offer guidance on what metrics should be seen as more or less appropriate for use in things like model selection, hyperparameter tuning, or being highlighted as the 'critical' metric in a given problem scenario.

**For context-independent model evaluation, use AUROC:** For model evaluations conducted outside of specific deployment contexts, where the differential costs of errors are undefined, the necessity for a metric that impartially values improvements across the entire model output space becomes paramount (Figure 1a). As it is not known in advance where samples of interest will live in the output space, nor are particular cost ratios known, there should be no preference for mistake correction. Therefore, in this setting, AUROC is favorable as it uniformly accounts for every correction, offering a fair assessment irrespective of decision thresholds.

**For deployment scenarios with elevated false negative costs, use AUROC:** In applications where the consequences of false negatives are especially high, such as in the early screening for critical illnesses like cancer (Figure 1c), a primary goal of the model will be to achieve the fewest missed cancer cases; equating to prioritizing model recall. In such a scenario, the most important mistakes to correct occur at lower score thresholds, as high-score mistakes will not change which positive samples are missed in deployment settings (as chosen thresholds are likely to be low). This behavior is the *inverse* of what AUPRC prioritizes, demonstrating that in such situations, AUROC should be preferred over AUPRC.

**For ethical resource distribution among diverse populations, use AUROC:** When faced with the challenge of ethically allocating scarce resources across a broad population, necessitating equitable benefit distribution among subgroups (Figure 1d), one must avoid prioritizing model improvements that selectively favor one subpopulation. As AUPRC will target high-score regions selectively, it risks unduely favoring high-prevalence subpopulations, as shown in Theorem 3 and Figures 2 and 3. Even though in this resource distribution problem, high-score regions are selectively important compared to low-score regions, the fact that in this problem, we must prioritize across all subpopulations equally means that AUPRC's global preference is untenable as it could induce bias.

**For reducing false positives in high-cost, single-group intervention prioritization or information retrieval settings, use AUPRC:** In scenarios where the cost associated with false positives significantly outweighs that of false negatives, absent of equity concerns—such as in selecting candidate molecules from a fragment library for drug development trials, where only the most promising molecules will proceed to costly experimental validation (Figure 1e)—the metric of choice should facilitate a reduction in high-score false positives. This necessitates a focus on correcting high-score errors, for which AUROC might not be ideal due to its uniform treatment of errors across the score spectrum, potentially obscuring improvements in critical high-stake decisions.

## 5  Literature Review: Examining how Claim 1 Became so Widespread

Claim 1 states that "AUPRC is better than AUROC in cases of class imbalance" and is widespread in the literature. Via both a manual literature search and an automated search of over 1.5M arXiv papers (see Appendix I for methodology), we observed 424 publications making this claim.[3] This widespread disemmination of Claim 1 *has a significant impact on the validity of scientific discourse*. We observed examples of high-profile papers operating in medically critical settings where high-recall is a key priority evaluating ML systems via AUPRC due to their task's underlying class imbalance [399]; papers focusing on fairness critical applications relying on AUPRC due to this claim, even while our results demonstrate AUPRC has major *problems* in the fairness regime [366, 306], and numerous other papers perpetuating this source of scientific misinformation.

Among the 424 papers we discovered referencing this claim, 167 did so with no associated citation. These papers were published in a wide range of venues, including high profile venues such as NeurIPS, ICML, and ICLR. This reflects not only the widespread belief in this claim, but also that we may be too comfortable making seemingly "correct" assertions without appropriate attribution in ML today. Further, Among the 257 that reference this claim and cite a source for this assertion, 135 *do not cite any papers that actually make this claim in the first place*. Most often, papers erroneously attribute this claim to [83], which was cited as a source for this claim 144 times. While [83] makes many interesting, meaningful claims about the ROC and PR curves, and *does argue that the precision-recall curve may be more informative than the ROC in cases of class imbalance* it never asserts that the *area under* the PR curve should be preferred over the *area under* the ROC in cases of class imbalance. This distinction is important, because while curves can be used to simultaneously communicate many different performance statistics to their viewers across different FPR/TPR or Precision/Recall criteria, and therefore should be assessed primarily as communication tools to deployment experts, *areas under these curves* are single-point summarizations which are primarily used for deployment-agnostic model comparison, which is a very different use case.

Even when appropriate papers are cited, the valid settings in which AUPRC should be preferred (see Section 4 for examples) are often over-shadowed by significant over-generalizations to preferring AUPRC in *all* settings featuring class imbalance. For example, claims such as that "precision-recall curves are more informative of deployment metrics" are often used to justify why AUPRC should be used in all cases of class imbalance, rather than just in cases where the relevant deployment metrics are most directly associated with the PR curve. Another class of arguments made in favor of Claim 1 is rooted in claims that AUROC is poor in cases of class imbalance because its scores are misleadingly high. While this argument can reflect a meaningful limitation of the communication value of the ROC or the AUROC, comments about singleton metric results (rather than model comparison through metric values) are inherently orthogonal to the goal of model evaluation. *In other words, what matters for model evaluation is not how high a given metric is, but rather the extent to which the metric meaningfully captures the right improvements in the model in the right ways.* The widespread nature of Claim 1 has also led researchers astray when exploring new optimization procedures for AUPRC, by advocating for the importance of AUPRC when processing skewed data, even in domains such as medical diagnoses that often have high false negative costs relative to false positive costs [409]. For a more extensive breakdown of the arguments we observed in the literature and the sources making them, see Appendix Tables 2 and 3.

---

[3]Note that throughout this section, full citations for all of the larger lists of papers we reference will be relegated to Appendix Section I for concision.

# 6 Limitations and Future Works

There are still a number of areas for further improvement and future work. Firstly, our theoretical findings can be refined and generalized to, e.g., take into account the difficulty of the target task (which may differ between subgroups), not require models to be calibrated (in the case of Theorem 3), or specifically take into account more than 2 subpopulations for more nuanced comparisons beyond what can be inferred through pairwise comparisons between subpopulations, where our results would naturally apply. Further, extending our real-world experiments to more fairness datasets and identifying more nuanced ways to probe the impact of metric choice on disparity measures would significantly strengthen this work. These analyses can also be extended to consider other metrics, such as the area under the precision-recall-gain curve [104], the area under the net benefit curve [384, 307], and single-threshold, deployment centric metrics as well. In addition, one of the largest limitations of Theorem 3 is its restrictive assumptions, in particular the requirement of perfect calibration. A ripe area of future work is thus to investigate how we can soften our analyses for models with imperfect calibration or to determine whether or not our results imply anything about the viability or safety of post-hoc calibration of models optimized either through AUPRC or AUROC.

# 7 Conclusion

This study interrogates the pervasive assumption within the ML community that AUPRC is a better evaluation metric than AUROC in class-imbalanced settings. Our empirical analyses and literature review reveal several concrete findings that challenge this notion. In particular, we show that while optimizing for AUROC equates to minimizing the model's FPR in an unbiased manner over positive sample scores, optimizing for AUPRC equates to minimizing the FPR specifically for regions where the model outputs higher scores relative to lower scores. *Further, we show both theoretically and empirically over synthetic and real-world fairness datasets that AUPRC can be an explicitly discriminatory metric through favoring higher-prevalence subgroups.*

In summary, our research advocates for a more thoughtful and context-aware approach to selecting evaluation metrics in machine learning. This paradigm shift, favoring a balanced and conscientious approach to metric selection, is essential in advancing the field towards developing not only technically sound, but also equitable and just models.

# Acknowledgements

MBAM gratefully acknowledges support from a Berkowitz Postdoctoral Fellowship. JG is funded by the National Institute of Health through DS-I Africa U54 TW012043-01 and Bridge2AI OT2OD032701.

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

# A Broader Impact and Ethical Considerations

This research paper challenges the conventional wisdom regarding the superiority of the AUPRC over AUROC in binary classification tasks with class imbalance and has several ethical implications and impacts.

Our analysis reveals that the preference for AUPRC in certain ML applications may not be empirically justified and could inadvertently amplify algorithmic biases. This calls for a re-examination of prevalent metrics within ML, especially in high-stakes domains like healthcare, finance, and criminal justice where biased models can have profound societal repercussions. The tendency of AUPRC to disproportionately favor models with higher prevalence of positive labels could exacerbate existing disparities, underscoring the ethical need for rigorous validation and scrutiny of evaluation metrics.

Additionally, our use of large language models for literature analysis demonstrates a novel approach in scrutinizing and re-evaluating long-standing assumptions in ML. This method could set a precedent for more comprehensive and robust scientific investigations in the field, fostering a culture of empirical rigor and ethical awareness.

The ethical dimension of our work lies in the spotlight it casts on metric selection in ML model evaluation. The potential of metrics like AUPRC to skew model performance favoring certain groups raises pressing concerns about fairness in algorithmic decision-making. This is particularly critical when algorithms influence key decisions affecting individuals and communities.

While we use the COMPAS dataset for recividism prediction in this work, we recognize the many societal issues with automated predictions of recidivism [93]. We utilize this dataset as it is a commonly used dataset in the fairness literature, but do not advocate for deployment of these models in any way.

Our study contributes to the technical discourse on metric behaviors in ML and serves as a cautionary tale against uncritically embracing established norms. It underscores the imperative for careful metric selection aligned with ethical principles and fairness objectives in ML, highlighting the far-reaching consequences of these choices in shaping societal outcomes and advancing the field of ML.

# B Code Availability

All code is available at `https://github.com/hzhang0/auc_bias` and `https://github.com/Lassehhansen/ArxivMLClaimSearch`.

# C Notation

Let $\mathcal{X}, \mathcal{Y} = 0, 1$ represent a paired feature and binary classification label space from which i.i.d. samples $(x, y) \in \mathcal{X} \times \mathcal{Y}$ are drawn via the joint distribution over the random variables $\mathsf{x}, \mathsf{y}$. Let $f_{\boldsymbol{\theta}} : \mathcal{X} \rightarrow (0, 1)$ be a binary classification model parametrized by $\boldsymbol{\theta} \in \mathbb{R}^d$ for some $d \in \mathbb{N}$ outputting continuous probability scores over this space.

We define random variable $\mathsf{s} = f_{\boldsymbol{\theta}}(\mathsf{x})$ to be the distribution of scores output by the model over input samples. Throughout the paper, $\boldsymbol{\theta}$ may be omitted if it is clear from context. We will occasionally also use the notation $\mathsf{s}_+$ and $\mathsf{s}_-$ to reflect the conditional distributions of model outputs conditioned on the label being 1 or 0, respectively:

$$\mathsf{s}_+ = f(\mathsf{x})|\mathsf{y} = 1$$
$$\mathsf{s}_- = f(\mathsf{x})|\mathsf{y} = 0.$$

Let $N_P$ be the number of data points with a positive label and $N_N$ the number with a negative label. Further, given a threshold $\tau$, define

$$\text{TP}_{\boldsymbol{\theta}}(\tau) = \left| \{ x_i \in \boldsymbol{X} | p_i^{(\boldsymbol{\theta})} \geq \tau, y_i = 1 \} \right|$$

$$\text{FN}_{\boldsymbol{\theta}}(\tau) = \left| \{ x_i \in \boldsymbol{X} | p_i^{(\boldsymbol{\theta})} < \tau, y_i = 1 \} \right|$$

$$\text{TN}_{\boldsymbol{\theta}}(\tau) = \left| \{ x_i \in \boldsymbol{X} | p_i^{(\boldsymbol{\theta})} < \tau, y_i = 0 \} \right|$$

$$\text{FP}_{\boldsymbol{\theta}}(\tau) = \left| \{ x_i \in \boldsymbol{X} | p_i^{(\boldsymbol{\theta})} \geq \tau, y_i = 0 \} \right|$$

$$\text{FR}(f, \tau) = P_{\mathsf{p}}(p > \tau)$$

$$\text{TPR}_{\boldsymbol{\theta}}(\tau) = \frac{\text{TP}_{\boldsymbol{\theta}}(\tau)}{\text{TP}_{\boldsymbol{\theta}}(\tau) + \text{FN}_{\boldsymbol{\theta}}(\tau)}$$

$$= P_{\mathsf{x}|\mathsf{y}=1}(f(x) > \tau)$$

$$= P_{\mathsf{s}|\mathsf{y}=1}(s > \tau)$$

$$= P(s_+ > \tau)$$

$$\text{FPR}_{\boldsymbol{\theta}}(\tau) = \frac{\text{FP}_{\boldsymbol{\theta}}(\tau)}{\text{FP}_{\boldsymbol{\theta}}(\tau) + \text{TN}_{\boldsymbol{\theta}}(\tau)}$$

$$= P_{\mathsf{x}|\mathsf{y}=0}(f(x) > \tau)$$

$$= P_{\mathsf{s}|\mathsf{y}=0}(s > \tau)$$

$$= P(s_- > \tau)$$

$$\text{Prec}_{\boldsymbol{\theta}}(\tau) = \frac{\text{TP}_{\boldsymbol{\theta}}(\tau)}{\text{TP}_{\boldsymbol{\theta}}(\tau) + \text{FP}_{\boldsymbol{\theta}}(\tau)}$$

$$= P_{\mathsf{y}|f(\mathsf{x})>\tau}(y = 1)$$

$$= P_{\mathsf{y}|\mathsf{s}>\tau}(y = 1)$$

Lastly, recall

$$\text{AUROC}_{\boldsymbol{\theta}} = \int_0^1 \text{TPR}_{\boldsymbol{\theta}} \frac{d\text{FPR}_{\boldsymbol{\theta}}}{d\tau} d\tau$$

$$= \int_0^1 \text{TPR}_{\boldsymbol{\theta}} d\text{FPR}_{\boldsymbol{\theta}}$$

$$= 1 - \int_0^1 \text{FPR}_{\boldsymbol{\theta}} d\text{TPR}_{\boldsymbol{\theta}}$$

$$\text{AUPRC}_{\boldsymbol{\theta}} = \int_0^1 \text{Prec}_{\boldsymbol{\theta}} \frac{d\text{TPR}_{\boldsymbol{\theta}}}{d\tau} d\tau$$

$$= \int_0^1 \text{Prec}_{\boldsymbol{\theta}} d\text{TPR}_{\boldsymbol{\theta}}$$

# D  Proof of Theorem 1

Recall that all notation is defined formally in Appendix C.

Here, we prove Theorem 1, which states

*Theorem* 1. Let $\mathcal{X}, \mathcal{Y} = \{0, 1\}$ represent a paired feature and binary classification label space from which i.i.d. samples $(x, y) \in \mathcal{X} \times \mathcal{Y}$ are drawn via the joint distribution over the random variables $\mathsf{x}, \mathsf{y}$. Let $f : \mathcal{X} \rightarrow (0, 1)$ be a binary classification model outputting continuous probability scores

over this space. Then,

$$\mathrm{AUROC}(f) = 1 - \mathbb{E}_{t \sim f(\mathsf{x}) | \mathsf{y}=1} \left[ \mathrm{FPR}(f,t) \right]$$

$$\mathrm{AUPRC}(f) = 1 - P_{\mathsf{y}}(y = 0) \mathbb{E}_{t \sim f(\mathsf{x}) | \mathsf{y}=1} \left[ \frac{\mathrm{FPR}(f,t)}{P_{\mathsf{x}}(f(x) > t)} \right]$$

*Proof.* Recall that AUROC and AUPRC are as follows:

$$\mathrm{AUROC} = \int_0^1 \mathrm{TPR} \, d\mathrm{FPR} = 1 - \int_0^1 \mathrm{FPR} \, d\mathrm{TPR}$$

$$\mathrm{AUPRC} = \int_0^1 \mathrm{Prec} \, d\mathrm{TPR}$$

However, we can further clarify these by leveraging the fact that $\mathrm{TPR}(\tau) = P_{\mathsf{s}_+}(s_+ > \tau) = \int_\tau^1 s_+(t)dt$, as below:

$$
\begin{aligned}
\int_0^1 g(\tau)d(\mathrm{TPR}(\tau)) &= \int_1^0 g(\tau) \frac{d\mathrm{TPR}(\tau)}{d\tau} d\tau \\
&= \int_1^0 g(\tau) \frac{d}{d\tau}(P_{\mathsf{s}_+}(s_+ > \tau)) d\tau \\
&= \int_1^0 g(\tau) \frac{d}{d\tau} \left( \int_\tau^1 s_+(t)dt \right) d\tau \\
&= \int_1^0 g(\tau)(-s_+(\tau)) d\tau \\
&= \mathbb{E}_{\mathsf{s}_+}[g]
\end{aligned}
$$

So, $\mathrm{AUROC} = 1 - \mathbb{E}_{\mathsf{s}_+}[\mathrm{FPR}]$ & $\mathrm{AUPRC} = \mathbb{E}_{\mathsf{s}_+}[\mathrm{Prec}]$. To further simplify, we expand $\mathrm{Prec}$ via Bayes rule:

$$
\begin{aligned}
\mathrm{Prec} &= 1 - P_{\mathsf{y}|\mathsf{s}>\tau}(y = 0) \\
&= 1 - \underbrace{P_{\mathsf{s}|\mathsf{y}=0}(s > \tau)}_{\mathrm{FPR}(\tau)} \frac{P_{\mathsf{y}}(y = 0)}{P_{\mathsf{s}}(s > \tau)}
\end{aligned}
$$

Thus,

$$
\begin{aligned}
\mathrm{AUROC}(f) &= 1 - \mathbb{E}_{t \sim \mathsf{s}_+}[\mathrm{FPR}(f,t)] \\
&= 1 - \mathbb{E}_{t \sim f(\mathsf{x}) | \mathsf{y}=1}[\mathrm{FPR}(f,t)] \\
\mathrm{AUPRC}(f) &= \mathbb{E}_{t \sim \mathsf{s}_+}[\mathrm{Prec}(f,t)] \\
&= 1 - P_{\mathsf{y}}(y = 0) \mathbb{E}_{t \sim s_+} \left[ \frac{\mathrm{FPR}(f,t)}{P_{s \sim \mathsf{s}}(s > t)} \right] \\
&= 1 - P_{\mathsf{y}}(y = 0) \mathbb{E}_{t \sim f(\mathsf{x}) | \mathsf{y}=1} \left[ \frac{\mathrm{FPR}(f,t)}{P_{\mathsf{x}}(f(x) > t)} \right]
\end{aligned}
$$

as desired. □

Synthetic validation of Theorem 1 can also be found in our public code. Note that this formulation of AUPRC reflects earlier, different formulations of AUPRC, such as those found in the AUPRC optimization literature [409].

# E   Proof of Theorem 2

Here, we prove Theorem 2, which states

*Theorem* 2. Define $f, \mathcal{X}, \boldsymbol{X}, \boldsymbol{y}$ and $N$ as in Definition 2.1. Further, suppose without loss of generality that the dataset $\boldsymbol{X}$ is ordered such that $f(x_i) < f(x_{i+1})$ for all $i$. Then, let us define $M = \{i | (x_i, x_{i+1})$ is an *incorrectly ranked adjacent pair* for model $f\}$. Define $f_i'$ to be a model that is identical to $f$ except that the probabilities assigned to $x_i$ and $x_{i+1}$ are swapped:

$$f_i' : x \mapsto \begin{cases} f(x) & \text{if } x \notin \{x_i, x_{i+1}\} \\ f(x_{i+1}) & \text{if } x = x_i \\ f(x_i) & \text{if } x = x_{i+1}. \end{cases}$$

Then, $\mathrm{AUROC}(f_i') = \mathrm{AUROC}(f_j')$ for all $i, j \in M$, and $\mathrm{AUPRC}(f_i') < \mathrm{AUPRC}(f_j')$ for all $i, j \in M$ such that $i < j$.

*Proof.* Suppose $f$ has a given, non-empty set $M$ of atomic mistakes, such that, without loss of generality, $(x_i, x_{i+1}) \in M$. Suppose we construct a new model $f'$ with empirical distributions $p_+'$ and $p_-'$ by replicating the scores assigned by the model $f$ with $x_i$ and $x_{i+1}$ swapped (i.e., we correct the mistake $(x_i, x_{i+1})$, so $x_i' = x_{i+1}$ and $x_{i+1}' = x_i$).

For which thresholds drawn from the original distribution $\mathsf{p}_+$ will the number of false positives of $f'$ differ from the number of false positives of $f$ at that same threshold? For any threshold $\tau < x_i$, fixing the mistake $(x_i, x_{i+1})$ will not change the number of false positives with threshold $\tau$, because both $x_i$ and $x_{i+1}$ are above $\tau$. For any threshold $\tau > x_{i+1}$, the number will likewise not change as both $x_i$ and $x_{i+1}$ are below $\tau$. The only $\tau$ that will have an impact is $\tau = x_i$ (recall that this is for an empirical distribution $p_+$ which contains $x_i$ and by the definition of atomic mistakes, there are no samples in $f$ with scores between $x_i$ and $x_{i+1}$). In $f$, the fact that $x_{i+1} > x_i$ yet has a negative label means that there will be one false positive corresponding to sample $i + 1$ greater than $x_i$ in addition to all those that exist with scores greater than $x_{i+1}$. For $f'$, however, the samples have swapped, so $x_i' > x_{i+1}'$ and thus there is no false positive corresponding to sample $i + 1$ at the positive score threshold corresponding to $x_i'$. Therefore, the number of false positives will only change to decrease by one for the threshold $x_i$ when the mistake $(x_i, x_{i+1})$ is corrected.

As AUROC weights the false positive rate at all positive samples equally and the false positive rate is proportional to the number of false positives, this shows that AUROC will improve by a constant amount no matter which atomic mistake is fixed. In contrast, as AUPRC weights false positives inversely by the model's firing rate, it will improve by an amount that is directly linearly correlated with the inverse of the model's firing rate, implying that it favors mistakes with higher scores and disfavors mistakes with lower scores.

Note that as we use strict inequalities in our definition of the decision rule underlying the FPR here, a pair of scores that are tied but have different labels will not induce a false positive at the corresponding positively labeled sample's threshold, so separating such ties will have no impact on AUROC whatsoever. It would similarly not impact AUPRC as neither the FPR nor the model firing rate will decrease when the negative sample within the tie is perturbed to be strictly below the positive sample. $\qquad\square$

Synthetic empirical validation of Theorem 2 can also be found in our public code.

# F   Proof of Theorem 3

In this section, we formally prove Theorem 2. We begin by establishing Lemma 1 and 2.

**Lemma 1.** *Let a model $f$ be perfectly calibrated and yield score distributions for positive and negative samples from probability density functions $p_+$ and $p_-$. Then $p_+(t) = \frac{t}{1-t} \frac{P_y(y=0)}{P_y(y=1)} p_-(t)$*

*Proof.* As this model is calibrated perfectly, we have that

$$p_+(t) = P_{\mathsf{s|y}=1}(s = t)$$

$$= \frac{P_{\mathsf{y|s}=t}(y = 1)p_{\mathsf{s}}(t)}{P_{\mathsf{y}}(y = 1)}$$

$$= t\frac{P_{\mathsf{y}}(y = 1)p_+(t) + P_{\mathsf{y}}(y = 0)p_-(t)}{P_{\mathsf{y}}(y = 1)}$$

$$= tp_+(t) + t\frac{P_{\mathsf{y}}(y = 0)}{P_{\mathsf{y}}(y = 1)}p_-(t).$$

Thus, $p_+(t) = \frac{t}{1-t}\frac{P_{\mathsf{y}}(y=0)}{P_{\mathsf{y}}(y=1)}p_-(t)$ as desired. $\qquad\square$

**Lemma 2.** *Let a model $f$ be perfectly calibrated and yield score distributions for positive and negative samples from probability density functions $p_+$ and $p_-$, with overall distribution given by $p(t) = P_{\mathsf{y}}(y = 1)p_+(t) + P_{\mathsf{y}}(y = 0)p_-(t)$. Then for all $\tau \in (0, 1)$, $\mathrm{FR}(f, \tau) \leq \frac{P_{\mathsf{y}}(y=1)}{\tau}$.*

*Proof.* By definition, we have

$$\mathrm{FR}(f, \tau) = \int_\tau^1 P_{\mathsf{y}}(y = 1)p_+(t) + P_{\mathsf{y}}(y = 0)p_-(t)dt$$

$$= \int_\tau^1 P_{\mathsf{y}}(y = 1)p_+(t) + P_{\mathsf{y}}(y = 1)\frac{1-t}{t}p_+(t)dt$$

$$= P_{\mathsf{y}}(y = 1)\int_\tau^1 \frac{1}{t}p_+(t)dt,$$

where step two leverages the fact that $f$ is perfectly calibrated and the result in Lemma 1.

As $t \geq \tau$, $\frac{1}{t} \leq \frac{1}{\tau}$. Then, as $p_+(t) \geq 0$, $\int_\tau^1 \frac{1}{t}p_+(t)dt \leq \frac{1}{\tau}\int_\tau^1 p_+(t)dt$. Finally, as $\int_0^1 p_+(t)dt = 1$, we see that $\int_\tau^1 p_+(t)dt \leq 1$. Therefore,

$$\mathrm{FR}(f, \tau) = P_{\mathsf{y}}(y = 1)\int_\tau^1 \frac{1}{t}p_+(t)dt$$

$$\leq P_{\mathsf{y}}(y = 1) \cdot \frac{1}{\tau} \cdot 1$$

$$= \frac{P_{\mathsf{y}}(y = 1)}{\tau}.$$

$\square$

*Theorem* 3. Let $f, \mathcal{X}, \boldsymbol{X}, \boldsymbol{y}, N, M$, and $f'_j$ all be defined as in Theorem 2. Further, suppose that in this setting the domain $\mathcal{X}$ now contains an attribute defining two subgroups, $\mathcal{A} = \{0, 1\}$, such that for any sample $(x_i, y_i)$, $a_i$ denotes the subgroup to which that sample belongs. Let $f$ be perfectly calibrated for samples in subgroup $a = 0$, such that $P_{\mathsf{y|a,x}}(y = 1|a = 0, f(x) = t) = t$. Then,

$$\lim_{P_{\mathsf{y|a}}(y=1|a=0)\to 0} P\left(a_i = a_{i+1} = 1 \middle| i = \arg\max_{j \in M}\left(\mathrm{AUPRC}(f'_j)\right)\right) = 1.$$

*Proof.* Given Theorem 2, the atomic mistake that would, upon correction, result in the largest improvement to AUPRC is the mistake which occurs at maximal score (as this minimizes the firing rate, which is the denominator in the weighting term for AUPRC). Suppose that at threshold $\tau$, the probability that a mistake will occur above score $\tau$ in subgroup 1 with $N$ samples drawn is at least $\delta \in (0, 1]$. As the parameters for subgroup 1 are fixed as we vary the prevalence for subgroup 2, $\tau$ can be seen as a constant with respect to the limit we are taking.

But, by Lemma 2 and by the fact that $f$ is perfectly calibrated for subgroup 2, we know that the probability that $f$ will output a score for sample 2 regardless of its label that exceeds $\tau$ is upper bounded by $\frac{p_{\mathsf{y}}^{(2)}}{\tau}$. In the limit as $p_{\mathsf{y}}^{(2)}$ tends to zero, the probability that any probabilities will be observed at our greater than $\tau$ from subgroup 2 likewise tends to zero.

This means that while the probability that we observe a mistake from subgroup 1 stays fixed at at least $\delta > 0$, the probability that we could observe any mistake that involves any sample from subgroup 2 (either a cross-group mistake or a purely subgroup 2 mistake) tends to zero, establishing the claim. $\qquad\square$

# G Details for Synthetic Experiments

## G.1 Sampling a random model with a given AUROC

A key component of our synthetic experiments is the ability to sample a set of model scores and labels randomly that will have a target AUROC. To do this, we use the following procedure (which may or may not be previously known; we derived it from scratch for this work, but make no claim about its novelty). Let $N$ be the number of points we are sampling overall, and $N_+$ be the number of positive points being sampled (which is dictated by the user given prevalence).

1. Uniformly sample a random collection of positive-label sample scores between zero and one.

2. Between each (ascending) model positive score indexed from 1 $p_+^{(i)}$ and $p_+^{(i+1)}$, we can count the number of positive samples that have scores less than any value in this window ($i$) and the number that have scores greater than any value in this window (which will be $N_+ - i$).

3. As the target AUROC is the probability that a randomly sampled negative will be ranked more highly than a randomly sampled positive, we can leverage the number of less-than positive scores $i$ and greater than positive scores $N_+ - i$ to compute the probability that a randomly sampled negative score will live in the window $(p_+^{(i)}, p_+^{(i+1)})$ via the binomial distribution.

4. Now, to sample a random negative, we simply first sample a random window $(p_+^{(i)}, p_+^{(i+1)})$ with the probabilities assigned above, then uniformly sample a value $p_-$ within that window. We can repeat this process to the target number of negative samples $N - N_+$ to form our final set of scores.

5. If desired, the output scores can further be scaled to have expectation given by the dataset's prevalence or can be adjusted via a calibration method to be calibrated given the assigned labels. Both procedures can be done without affecting the AUROC. Note that as any calibrated model will have expected probability given by the label's prevalence (See Appendix G.2), the former condition is strictly weaker than the latter.

The procedure outlined above guarantees that, in expectation, the AUROC of the generated set of scores and labels will be precisely the target AUROC. However, if you apply this procedure indpendently across different sample subpopulations, this guarantee can only be applied on each subpopulation individually, and not necessarily on the overall population due to the unspecified xAUC term. However, in practice, for the experiments we ran here, that impact neither meaningfully impacts our experiments nor were the joint AUROCs sufficiently different from the target AUROC to warrant a more complex methodology.

## G.2 Calibration includes prevalence matching

Let $\mathsf{p}$ be a random variable describing the probabilities (not scores) output by the model over the input distribution defined by the data generative function. If a model is calibrated, this means that $P_{\mathsf{y}|\mathsf{p}}(y = 1|p = q) = q$ — that the probability that the label for a given point is 1 is given precisely by the models output probability for that sample. With that in mind, we have:

$$\mathbb{E}_{\mathsf{p}}[q] = \mathbb{E}_{\mathsf{p}}\left[P_{\mathsf{y}|\mathsf{p}}(y = 1|p = q)\right]$$
$$= \int_0^1 P_{\mathsf{y}|\mathsf{p}}(y = 1|p = q)p_{\mathsf{p}}(q)dq$$
$$= \int_0^1 P_{\mathsf{y},\mathsf{p}}(y = 1, p = q)dq$$
$$= P_{\mathsf{y}}(y = 1)$$

## G.3 Details on optimization procedures

M1. **Adding Random Noise.** We sample a vector $\epsilon \in \mathbb{R}^n$, where each element is uniformly drawn from $[-\delta, \delta]$. We compute the selection metric for $S' = S + \epsilon$. We repeat this

procedure 100 times, and return the $S'$ that achieves the maximum value for the selection metric. We vary the maximum magnitude of the perturbation $\delta \in [0, 0.1]$ in a grid. Results for this setting are shown in Figure 6.

We note that this approach is subtly biased in favor of the lower-prevalence group. In particular, because scores for the low-prevalence group tend to be "squished" into a smaller region of the probability space, a random perturbation of fixed magnitude will proportionally induce more score *permutations* in the low-prevalence group than the high-prevalence group, which affords the system greater capacity to improve the model for the low-prevalence group independent of the choice of AUROC or AUPRC.

M2. **Sequentially Fixing Atomic Mistakes.** We sequentially correct atomic mistakes, as defined in Figure 1. At each step, we first discover the set of all atomic mistakes $M$. To maximize AUROC, we randomly select a pair $(S_i, S_j) \in M$, and swap their scores in $S$, i.e. $S'_i = S_j, S'_j = S_i$. To maximize AUPRC, we swap the scores for the pair $(S_i, S_j) = \arg\max_{(s_i, s_j) \in M} s_j$. We repeat this process for 50 steps, with each one sequentially fixing another atomic mistake in $S$. Results for this setting are shown in Figures 4b and 4a.

M3. **Sequentially Permuting Nearby Scores.** We first sort $S$ and $Y$ such that $S$ is in ascending order. We apply a random permutation to $S$ by re-indexing it using a random ordering, but such that scores are not shuffled too far from their original index. Let $\sigma$ be the ordered sequence $(1, 2, ..., n)$. Define $\Omega$ to be the set of all permutations of $\sigma$, such that for all $\omega \in \Omega$, $|\omega_i - \sigma_i| \leq \gamma$ for $i \in \{1, ..., n\}$. At each step, we sample $\omega \in \Omega$ with $\gamma = 3$ twenty times, where each $\omega$ corresponds to a new candidate ordering of $S$. We compute the selection metric for each of the twenty orderings, and return $S'$ to be the score permutation that achieves the maximum value for the selection metric. We repeat this procedure for 25 steps, setting $S$ at each step to be the $S'$ output from the previous step. Results for this setting are shown in Figures 4d and 4c.

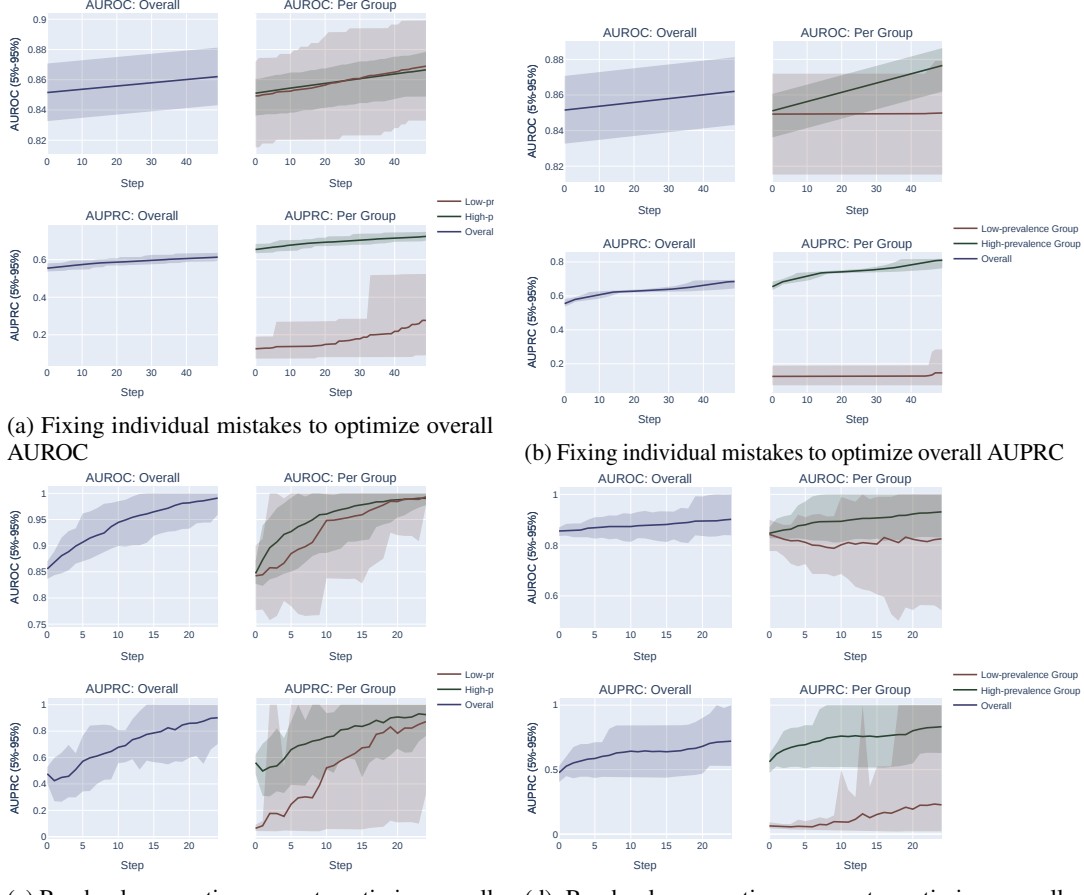

(a) Fixing individual mistakes to optimize overall AUROC

(b) Fixing individual mistakes to optimize overall AUPRC

(c) Randomly permuting scores to optimize overall AUROC

(d) Randomly permuting scores to optimize overall AUPRC

Figure 4: Comparison of the impact of optimizing for overall AUROC and overall AUPRC on the per-group AUROC and AUPRCs of two groups in a synthetic setting, using both the *sequentially fixing individual mistakes* optimization procedure (M2; *top*) and the *sequentially permuting nearby scores* optimization procedure (M3; *bottom*) described in Section 3.1. Note that the prevalence of $Y$ in the high-prevalence group and the low-prevalence group are 0.05 and 0.01 respectively.

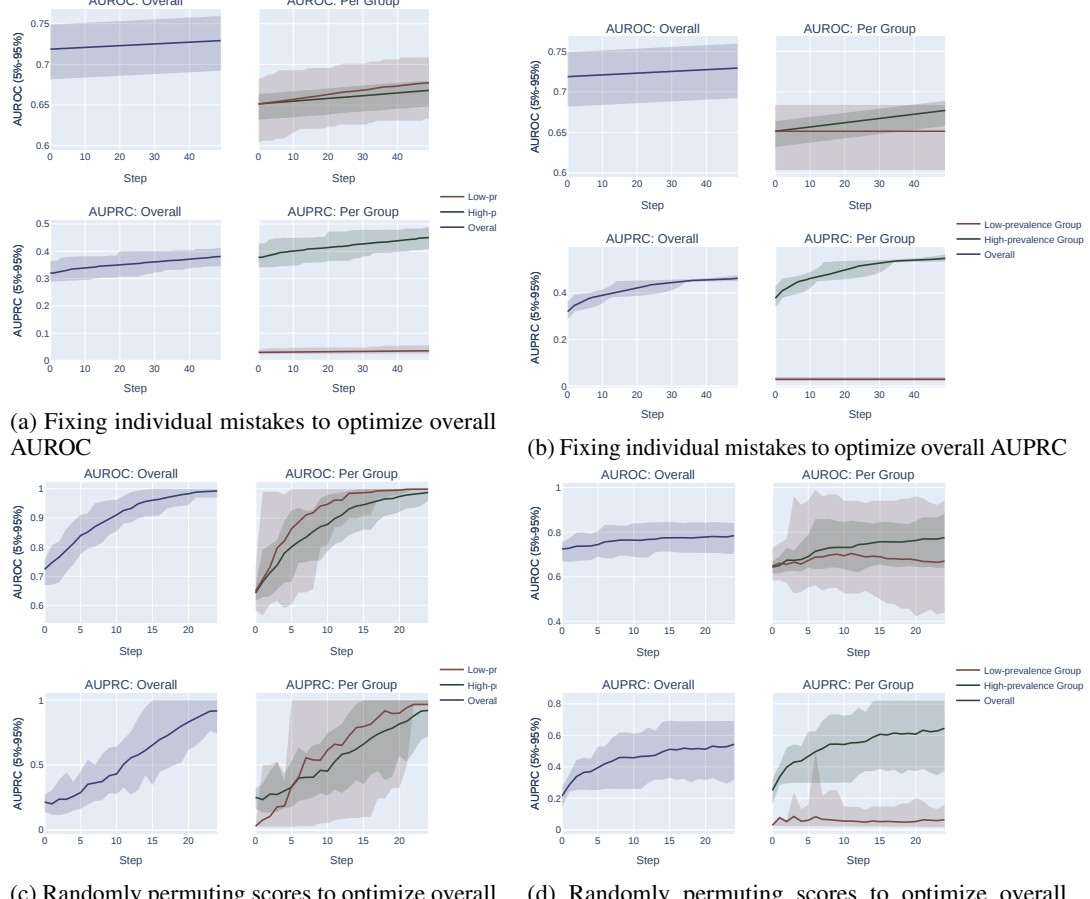

(a) Fixing individual mistakes to optimize overall AUROC

(b) Fixing individual mistakes to optimize overall AUPRC

(c) Randomly permuting scores to optimize overall AUROC

(d) Randomly permuting scores to optimize overall AUPRC

Figure 5: Comparison of the impact of optimizing for overall AUROC and overall AUPRC on the per-group AUROC and AUPRCs of two groups in a synthetic setting where the initial AUROC was set to 0.65 rather than 0.85, using both the *sequentially fixing individual mistakes* optimization procedure (M2; *top*) and the *sequentially permuting nearby scores* optimization procedure (M3; *bottom*) described in Section 3.1. Note that the prevalence of $Y$ in the high-prevalence group and the low-prevalence group are 0.05 and 0.01 respectively.

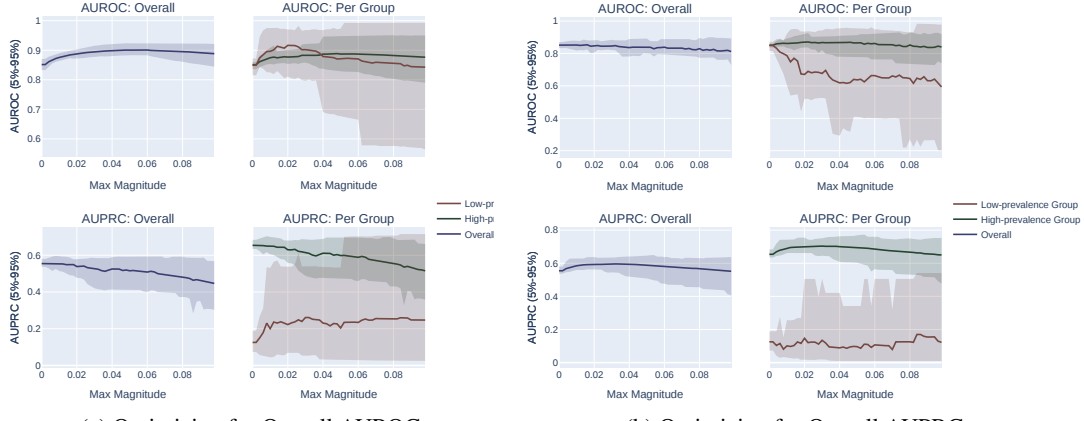

Figure 6: Comparison of the impact of optimizing for overall AUROC and overall AUPRC on the per-group AUROC and AUPRCs of two groups in a synthetic setting, using the *adding random noise* optimization procedure (M1) described in Section 3.1. Note that the prevalence of $Y$ in $G_1$ and $G_2$ are 0.05 and 0.01 respectively.

# H   Additional Details on Real World Experiments

## H.1   Dataset Details

We use the following four datasets. In all datasets, we use sex and race as protected attributes.

- `adult` [17]: The UCI Adult dataset, where the goal is to predict whether an individual's income is $> \$50k$.
- `compas` [14]: The task to predict two-year recidivism. We only select samples belonging to "African-American" and "Caucasian", leading to a binary race variable.
- `lsac` [413]: The task is to predict whether a law school applicant will pass the bar. We only select samples belonging to White and Black applicants.
- `mimic` [178]: We use the in-hospital mortality task proposed by [138], where the goal is to predict whether a patient will die in the ICU given labs and vitals from the first 48 hours of their hospital stay. We only select samples belonging to White and Black patients.

In each dataset, we balance the groups by subsampling the majority group. We then split each dataset into 50% training, 25% validation, 25% test sets, stratified by the group. Dataset statistics can be found in Table 1.

Table 1: Dataset statistics for the four binary classification datasets used in this study. Note that $n$ refers to the number of samples *after* balancing by the corresponding attribute. Here, "Prevalence (Higher)" refers to the rate at which the prediction label $y = 1$ for the subpopulation with a higher such rate, and "Prevalence (Lower)" refers to the same rate but over the subpopulation of the dataset with a lower rate of $y = 1$.

| Dataset | Attribute | n | # Features | Prevalence (Higher) | Prevalence (Lower) |
|---------|-----------|------|-----------|--------------------|--------------------|
| adult   | Sex       | 20,394 | 12 | 30.1% | 10.7% |
| adult   | Race      | 6,248  | 12 | 24.6% | 12.2% |
| compas  | Sex       | 2,438  | 6  | 52.0% | 39.4% |
| compas  | Race      | 4,908  | 6  | 55.4% | 42.3% |
| lsac    | Sex       | 15,906 | 8  | 96.0% | 95.0% |
| lsac    | Race      | 2,396  | 8  | 96.5% | 77.2% |
| mimic   | Sex       | 15,632 | 49 | 12.5% | 11.9% |
| mimic   | Race      | 4,030  | 49 | 13.9% | 9.3% |

## H.2   Hyperparameter Grid

We use the following hyperparameter grid for our experiments:

- max depth: {1, 2, ..., 9}
- learning rate: [0,01, 0.3]
- number of estimators: [50, 1000]
- min child weight: {1, 2, ..., 9}
- use protected attribute as input feature: {yes, no}
- group weight of higher prevalence group: {1, 2, 3, 4, 5, 10, 15, 20, 25, 50}

## H.3   Additional Results

All raw AUC results can be found in the code repository for these experiments, at `https://github.com/hzhang0/auc_bias`. Additionally, summarized results in different views can be found in Figures 7 and 8.

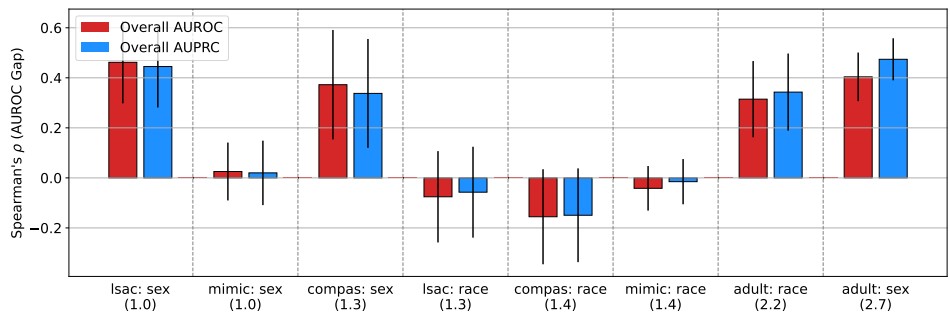

Figure 7: Spearman's $\rho$ between the test-set signed AUROC gap versus the validation set overall AUPRC, and the AUROC gap versus the overall AUROC. Numbers in parentheses are the prevalence ratios between the two groups for the particular attribute, and datasets are sorted by this quantity. Error bars are 95% confidence intervals from 20 different random data splits.

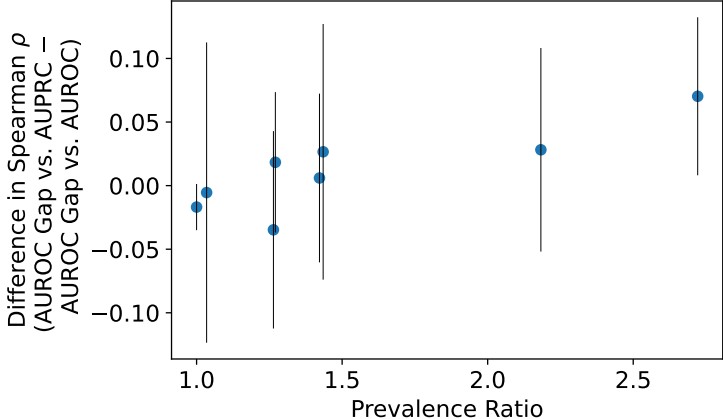

Figure 8: Correlation between the prevalence ratio, and the difference between the Spearman's $\rho$ of the AUROC gap versus AUROC and the AUROC gap versus AUPRC. Each point represents a dataset and attribute combination. *This correlation itself has a Spearman's $\rho$ of $0.905$ (p = 0.002).*

# I   Literature Review Methodology

## I.1   Paper Acquisition

The initial phase of our comprehensive literature search involved the acquisition of datasets from both the ArXiv preprint server (through the RedPajama dataset on Hugging Face), as well as from a subset of years of NeurIPS, ICML, ICLR, ACL, and CVPR conference proceedings (all scraped manually). The ArXiv dataset, approximately 93.8 GB in size, encompassed over 1.5 million texts in JSONL format. For NeurIPS, we developed a script to scrape conference papers from 1987 to 2019 (9680 texts), aiming to enrich our search. Other venues contributed fewer papers to our assessment process.

## I.2   Keyword-Driven Filtering Process

1. **Keyword List Development:** We developed two distinct keyword lists to systematically identify papers relevant to our research on AUROC (Area Under the Receiver Operating Characteristic) and AUPRC (Area Under the Precision-Recall Curve) in our initial screening phase. The keyword lists can be accessed here for AUPRC and here for AUROC.

2. **Automated Script-Based Search:** Python scripts were employed to traverse the Arxiv and NeurIPS datasets. These scripts detected occurrences of our predefined keywords, allowing efficient parsing of a vast number of texts from both sources.

3. **Dual Mention Selection Criterion:** We focused on papers discussing both AUROC and AUPRC. This criterion ensured the relevance of the papers to our research question. Through

this process, we narrowed the pool from 16,022 texts (containing either set of keywords) to 8,244 texts mentioning both in the Arxiv dataset. In the NeurIPS dataset, out of 9,680 texts reviewed, 78 were found to contain keywords from AUPRC and AUROC.

## I.3 AI-Assisted Screening and Refinement

1. **Preliminary Analysis with GPT-3.5:** We utilized OpenAI's GPT-3.5 model for an initial round of AI-assisted analysis for the arXiv dataset. This model identified and extracted papers making explicit claims regarding the comparative effectiveness of AUPRC over AUROC in scenarios of class imbalance, reducing our dataset from Arxiv to 2,728 papers.

2. **Further Refinement Using GPT-4.0 Turbo:** To refine our dataset further, we employed the GPT-4.0 Turbo model. Out of the 2,728 papers scrutinized from Arxiv using this model, 201 were found to be relevant. For NeurIPS, our focused search with GPT-4 resulted in identifying 2 papers of particular relevance to our thesis from the initial set that contained keywords related to both AUPRC and AUROC.

## I.4 Manual Review

- **Shared Document for Collaborative Analysis:** We compiled all pertinent papers, along with their respective Arxiv IDs and the claims identified by GPT-4.0 Turbo, into a shared Google document for team review. Claims made in papers were found manually, and the specific quote of the claim they made was highlighted along with whether or not they had a citation for this claim.

### I.4.1 Final Papers

After manual review, we identified 424 papers that make or reference some version of the claim that "AUPRC is better than AUROC in cases of class imbalance." [78, 212, 428, 116, 8, 223, 229, 318, 393, 308, 450, 407, 4, 217, 391, 417, 261, 282, 70, 399, 169, 343, 153, 148, 361, 281, 219, 206, 204, 2, 350, 156, 134, 425, 22, 115, 257, 429, 353, 279, 85, 424, 137, 258, 367, 10, 454, 251, 275, 140, 208, 438, 356, 79, 52, 444, 90, 252, 111, 296, 325, 293, 312, 321, 387, 358, 41, 397, 54, 159, 369, 71, 180, 301, 74, 363, 24, 86, 49, 253, 270, 390, 329, 333, 348, 238, 233, 6, 124, 448, 359, 236, 243, 335, 210, 58, 317, 284, 21, 192, 12, 379, 274, 202, 415, 377, 373, 271, 82, 193, 376, 242, 183, 88, 276, 322, 395, 327, 239, 338, 319, 149, 288, 408, 136, 128, 46, 163, 132, 28, 207, 411, 302, 412, 227, 101, 221, 145, 328, 277, 234, 73, 60, 418, 92, 20, 198, 103, 27, 410, 151, 174, 29, 287, 11, 304, 347, 161, 285, 437, 341, 366, 5, 225, 334, 423, 297, 47, 324, 147, 371, 38, 323, 53, 184, 344, 378, 162, 394, 414, 106, 110, 426, 346, 130, 264, 432, 179, 440, 191, 94, 189, 362, 316, 220, 175, 421, 39, 188, 19, 401, 389, 419, 309, 224, 109, 95, 197, 97, 31, 244, 135, 351, 133, 352, 402, 63, 405, 23, 81, 305, 396, 330, 266, 303, 294, 43, 62, 364, 295, 168, 181, 165, 114, 89, 66, 250, 248, 278, 13, 430, 25, 416, 34, 398, 199, 84, 56, 381, 76, 35, 269, 291, 404, 228, 185, 241, 453, 211, 160, 15, 209, 173, 420, 326, 286, 64, 273, 9, 299, 382, 139, 442, 96, 372, 201, 30, 446, 386, 403, 127, 196, 69, 155, 75, 360, 231, 235, 310, 267, 255, 345, 349, 171, 126, 190, 452, 342, 57, 182, 59, 455, 439, 123, 383, 112, 59, 357, 55, 142, 214, 203, 374, 314, 170, 87, 187, 300, 355, 262, 7, 260, 268, 280, 1, 44, 298, 265, 120, 249, 91, 422, 436, 18, 108, 113, 176, 283, 245, 433, 158, 218, 122, 313, 172, 380, 365, 154, 385, 290, 388, 247, 216, 67, 98, 368, 177, 164, 289, 320, 32, 186, 400, 16, 237, 42, 431, 157, 375, 272, 406, 259, 315, 117, 77, 152, 306, 434, 72, 166, 392, 449, 230, 246, 26, 445, 232, 146, 118, 107, 61, 129, 50, 143, 427, 144, 370, 311, 336, 447, 105, 263, 256, 215, 125, 167, 240, 226, 195, 48, 194, 200, 3, 213, 354, 340, 45, 141, 102, 332, 51, 443, 33, 68]**.**

Those papers that reference this claim without citation include [78, 212, 428, 116, 8, 223, 407, 4, 217, 261, 70, 169, 343, 153, 219, 206, 204, 2, 350, 156, 134, 22, 115, 257, 353, 279, 85, 424, 258, 367, 10, 454, 251, 275, 140, 356, 79, 52, 444, 90, 252, 111, 296, 293, 321, 358, 41, 54, 159, 369, 71, 180, 301, 74, 363, 24, 86, 253, 270, 390, 329, 333, 348, 233, 6, 124, 448, 359, 236, 243, 335, 210, 284, 21, 12, 379, 274, 202, 415, 377, 373, 271, 82, 376, 242, 183, 88, 322, 395, 239, 338, 149, 288, 408, 136, 128, 46, 163, 28, 412, 101, 328, 73, 60, 20, 103, 151, 29, 287, 11, 304, 161, 285, 341, 5, 423, 324, 371, 53, 184, 344, 162, 414, 110, 346, 264, 179, 191, 94, 189, 362, 316, 175, 188, 19, 389, 419, 109, 95, 351, 133, 23, 396, 303, 43, 364, 168, 165, 114, 250, 248, 430, 25, 416, 34, 398, 199, 56, 76, 35, 269, 291, 404, 228, 241, 453, 211, 160, 15, 420, 286, 9, 299, 139, 442, 96, 372, 30, 196, 69, 75, 360, 231, 235, 267, 171, 190, 342, 57, 439, 383, 112, 59, 357, 214, 314, 87, 187, 300, 262, 7, 260, 280, 44, 120, 249, 91, 18, 108, 113, 176, 283, 245, 122, 172, 380, 154, 385, 290, 216, 67, 164, 289, 186, 400, 16, 42, 431, 157, 375, 77, 306, 166, 449, 230, 232, 146, 118, 107, 61, 370, 311, 336, 447, 105, 263, 256, 125, 240, 3, 354, 340, 45, 141, 102, 332, 443]

Those that do so while citing only other papers that themselves never reference or argue Claim 1 include [428, 219, 204, 350, 156, 134, 22, 454, 251, 275, 356, 79, 52, 444, 90, 358, 41, 159, 369, 74, 86, 270, 333, 6, 124, 359, 243, 210, 284, 12, 379, 415, 377, 373, 395, 239, 136, 128, 328, 73, 20, 103, 29, 11, 304, 285, 5, 162, 414, 264, 191, 94, 189, 175, 419, 133, 303, 364, 165, 250, 25, 416, 398, 35, 228, 241, 453, 211, 160, 15, 286, 139, 96, 372, 30, 69, 75, 360, 231, 235, 171, 112, 59, 357, 314, 262, 7, 44, 120, 18, 113, 176, 283, 122, 172, 154, 385, 290, 216, 289, 431, 157, 375, 230, 146, 118, 107, 61, 311, 336, 3, 45, 141, 332]

All papers identified, manual screening results, and extracted quotes will be made available upon publication.

| Claim | References | Commentary |
|---|---|---|
| Precision-recall curves or other associated metrics *may* more appropriately reflect deployment objectives than the receiver operating characteristic. | [78, 212, 428, 279, 85, 137, 6] | While this claim is true, the informativeness of the PR curve for target deployment metrics is not sufficient to conclude that the AUPRC is superior to the AUROC in all cases of class imbalance. Despite this, it is often taken to assert this more general claim without caveat. |
| AUPRC does not depend on the number of true-negatives, so will be less optimistic than the AUROC | [212, 204, 2, 258, 275, 356, 79, 90, 325, 159, 180, 301, 329, 395] | As shown in Theorem 1, AUROC and AUPRC can both be naturally expressed as a function of the expectation of the model's false positive rate. More generally, lack of dependence on one quadrant among the mutually dependent four quadrants of a confusion matrix is not an informative property for the AUROC and AUPRC metrics. |
| AUPRC will often be significantly lower, farther from optimality, and/or will grow more non-linearly as model performance improves than AUROC for low-prevalence tasks | [212, 428, 257, 79] | Metric utility for model comparison depends on how appropriately it prioritizes model improvements, and is therefore less about the raw magnitude of the metric and more about the situations in which the order of a set of models will differ under one metric vs. another. One could easily make AUROC yield smaller values or grow more quickly near optimality by simply exponentiating it, but this would not yield a better metric. |
| AUPRC depends on prevalence, which is a desirable property | [282] | This statement is too vague to be formally evaluated; whether or not this dependence on prevalence is desirable depends on the context. For model comparison in general, we argue it is not desirable in this form as it induces the biases inhere in AUPRC previously discussed. |
| AUPRC better captures differentiating a positive sample with high score from a "hard" negative sample ("hard" meaning one also with high score) | [193] | While this claim is true by Theorem 2, it is not clear why this would be desired in general; this implicitly favors comparing "hard" negatives against "easy" positives as opposed to "easy" negatives against "hard" positives. |
| AUROC is otherwise "optimistic" in low-prevalence settings | [78, 4, 417, 261, 70, 134, 425, 251, 363, 238, 6, 448, 236, 335, 58, 317, 12, 202, 183, 149] | This claim is underspecified, and un-true. AUROC always means the same thing, probabilistically, and that meaning independent from class imbalance. |
| AUPRC focuses more on the positive (minority) class | [393, 417, 361, 281, 22, 115, 85, 10, 275, 356, 296, 293, 82, 193, 239, 319, 288] | This is unfounded; both AUROC and AUPRC are weighted expectations over the model's false positive rate—AUPRC cares more about samples in regions of low firing rate, not explicitly about positive or minority samples. |
| AUROC can not appropriately detect models with poor recall | [282] | This claim is unfounded; the AUROC clearly depends on the model's recall. Besides, if recall is the measure of interest, then that should be measured explicitly. |

Table 2: Various arguments and our responses to them present on a subset of papers for this claim in the literature.

## I.5   Code Availability

All code pertaining to the literature review search can be found in the following GitHub repository: `https://github.com/Lassehhansen/ArxivMLClaimSearch`

| Claim | References | Valid? | Commentary |
|---|---|---|---|
| Precision-recall curves or other associated metrics *may* more appropriately reflect deployment objectives than the receiver operating characteristic. | [78, 212, 339, 435, 40, 292, 331, 451, 222, 428] | ✓ | While this claim is true, the informativeness of the PR curve for target deployment metrics is insufficient to conclude that the AUPRC is superior to the AUROC in all cases of class imbalance. Despite this, it is often taken to assert this more general claim without caveat. |
| AUPRC does not depend on the number of true negatives, so will be less optimistic than the AUROC | [212, 121, 79] | | As shown in Theorem 1, AUROC and AUPRC can both be naturally expressed as a function of the expectation of the model's false positive rate. More generally, the lack of dependence on one quadrant among the mutually dependent four quadrants of a confusion matrix is not an informative property for the AUROC and AUPRC metrics. |
| AUPRC will often be significantly lower, farther from optimality, and/or will grow more non-linearly as model performance improves than AUROC for low-prevalence tasks | [212, 435, 121, 254, 331, 451, 222, 428, 79] | ✓ | Metric utility for model comparison depends on how appropriately it prioritizes model improvements. Therefore, it is less about the raw magnitude of the metric and more about the situations in which the order of a set of models will differ under one metric vs. another. One could easily make AUROC yield smaller values or grow more quickly near optimality by simply exponentiating it, but this would not yield a better metric. |
| AUPRC depends on prevalence, which is a desirable property | [339, 121, 435] | | This statement is too vague to be formally evaluated; whether or not this dependence on prevalence is desirable depends on the context. For model comparison in general, we argue it is not desirable in this form as it induces the biases in AUPRC previously discussed. |
| AUPRC better captures differentiating a positive sample with high score from a "hard" negative sample ("hard" meaning one also with high score) | [331] | ✓ | While this claim is true by Theorem 2, it is not clear why this would be desired in general; this implicitly favors comparing "hard" negatives against "easy" positives as opposed to "easy" negatives against "hard" positives. |

Table 3: Various arguments and our responses to them present on a subset of papers for this claim in the literature.

