# OpenReview forum: "A Closer Look at AUROC and AUPRC under Class Imbalance"
_NeurIPS.cc/2024/Conference — NeurIPS 2024 poster_

### Official Review · Reviewer_pyKS · 2024-06-15

**Soundness:** 4
**Presentation:** 2
**Contribution:** 3
**Rating:** 5
**Confidence:** 4

**Summary:**

A widespread claim in machine learning that AUPRC is a superior metric than AUCROC for tasks with class imbalance is not strictly true. Based on this statement, this paper challenges this claim from two perspectives. On the one hand, the authors theoretically characterize the behavior of AUROC and AUPRC in the presence of model mistakes, showing that optimizing for AUROC equates to minimizing the model’s FPR in an unweighted manner, whereas optimizing for AUPRC equates to minimizing the FPR in a weighted manner. On the other hand, experiments on both semi-synthetic and real-world fairness datasets support their theory.

**Strengths:**

-	The proposed ideas are novel and innovative. Theoretically, the authors explore the relationship between AUROC and AUPRC, revealing their key differences. Specifically, the authors show that while optimizing for AUROC equates to minimizing the model’s FPR in an unbiased manner over positive sample scores, optimizing for AUPRC equates to reducing the specifically for regions where the model outputs higher scores relative to lower scores. In addition, the authors propose that AUPRC is explicitly discriminatory in favor of high-scoring subpopulations.
-	The authors validate the proposed theoretical findings with the help of numerical results. To rigorously confirm the findings of differences between AUROC and AUPRC, the authors conduct many synthetic experiments and real-world validation on popular public fairness datasets.
-	Guidance on how to choose AUROC and AUPRC is provided. In section 4, the authors offer detailed instructions for use in different scenarios. For context-independent model evaluation, deployment scenarios with elevated false negative cost, and ethical resource distribution among diverse populations, AUCROC is a more proper metric. However, for reducing false positives in high-cost, single-group intervention prioritization or information retrieval settings, AUPRC is a better choice.

**Weaknesses:**

- The presentation needs to improve. For example, the term “high prevalence subgroup” is introduced without explanation, hindering my understanding of the theorem. A more detailed explanation of the high and low-prevalence subgroups should be provided. Besides, In theorem 3, the authors say that there exists a prevalence disparity sufficiently severe, but I can't see it directly from this theorem and suggest that the authors provide a clearer explanation.
- I find this paper (Exploring the Algorithm-Dependent Generalization of AUPRC Optimization with List Stability, NeurIPS 2022) also has a similar form (equation 2) and conclusion about AUCPRC. Can you explain the difference between your method and this one? The corresponding citation is also necessary.
- The caption of Figure 1 is too long. It is recommended to make it more concise.

**Questions:**

- Theorem 3 is based on two subgroups but real-world datasets generally have more than subgroups. Is recommended that the authors provide analysis based on multiple subgroups.

**Limitations:**

Yes

---

> ### Author Rebuttal · Authors · 2024-08-07
>
> Thank you for your apt and constructive review!
>
> ### W1: Can the presentation be improved, particularly regarding the clarity of "prevalence"?
> In essence, in Theorem 3, we show that if one group of samples has a much higher outcome rate (prevalence) than another (e.g., men are more likely than women to receive a correct diagnosis for a heart attack), optimizing for AUPRC will provably preferentially optimize to fix mistakes that affect the high prevalence group over the low prevalence group (e.g., the model will preferentially learn to identify heart attack symptoms for men at the expense of learning those for women). In settings where such a model is used to determine who should receive some limited resource (e.g., to be evaluated by a cardiac specialist after an ED visit), this translates into a disparity in resource allocation between the two groups, which in many cases may be undesirable.
>
> To clarify this in our text, we have added: _"Essentially, Theorem 3 (proof provided in Appendix F) shows the following. Suppose we are training a model $f$ over a dataset with two subpopulations: Population $a=0$ and $a=1$. If the model $f$ is calibrated and the rate at which $y=1$ for population $a=0$ is sufficiently low relative to the rate at which $y=1$ for population $a=1$, then the mistake that, were it fixed, would maximally improve the AUPRC of $f$ will be a mistake purely in population $a=1$. This demonstrates that AUPRC provably favors higher prevalence subpopulations (those with a higher base rate at which $y=1$) under sufficiently severe prevalence imbalance between subpopulations."_
>
> This clarifies that prevalence relates to the probability of $y=1$ for a specific subgroup. The prevalence disparity is captured by the limit as $P(y=1|a=0)$ approaches zero while $P(y=1|a=1)$ remains fixed. If you have further feedback on how to further clarify this point, please let us know.
>
>
> ### W2: How does your work relate to the NeurIPS 2022 paper on AUPRC optimization?
> Thank you for pointing out this related work! We've included it as a reference and explained how our findings are synergistic with and extend upon this excellent prior work:
>   1. In our proof of Theorem 1: "_Note that this formulation of AUPRC reflects earlier, different formulations of AUPRC, such as those found in the AUPRC optimization literature (Wen et al., 2022)_"
>   2. In our Synthetic Results section, where we comment on the impact of optimizing for AUROC vs. AUPRC: "_These results demonstrate explicitly that not only does optimizing for AUPRC differ greatly than for AUROC, as has been noted historically by researchers developing explicit AUPRC optimization schemes (Wen et al., 2022), but it in fact does in an explicitly discriminatory way in very realistic scenarios._"
>   3. Finally, in Section 5 (our literature review), we note that "_The widespread nature of Claim 1 has also led researchers astray when exploring new optimization procedures for AUPRC, by advocating for the importance of AUPRC when processing skewed data, even in domains such as medical diagnoses that often have high false negative costs relative to false positive costs (Wen et al., 2022)._""
>
> To further clarify the novelty of our work relative to that of Wen et al., note that while Wen et al. provide a related probabilistic expression for AUPRC designed to facilitate their optimization algorithm, our Theorem 1 (which presents a different formulation of AUPRC) is designed to clearly show the mathematical relationship between AUROC and AUPRC. Our impact comes from realizing the implications of this relationship on the strengths and weaknesses of these metrics, challenging the popular opinion that class imbalance is the defining factor in their distinct use cases.
>
>
> ### W3: Can you shorten the caption of Figure 1?
> We've shortened the caption considerably; it now reads: _"a) Consider a model $f$ yielding continuous output scores for a binary classification task applied to a dataset consisting of two distinct subpopulations, $\mathcal{A} \in \{0, 1\}$. If we order samples in ascending order of output score, each misordered pair of samples (e.g., mistakes 1-4) represents an opportunity for model improvement. Theorem 3 shows that a model's AUROC will improve by the same amount no matter which mistake you fix, while the model's AUPRC will improve by an amount correlated with the score of the sample. b) When comparing models absent a specific deployment scenario, we have no reason to value improving one mistake over another, and model evaluation metrics should therefore improve equally regardless of which mistake is corrected. c) When false negatives have a high cost relative to false positives, evaluation metrics should favor mistakes that have *lower scores*, regardless of any class imbalance. d) When limited resources will be distributed among a population according to model score, *in a manner that requires certain subpopulations to all be offered commensurate possible benefit from the intervention for ethical reasons*, evaluation metrics should prioritize the importance of within-group, high-score mistakes such that the highest risk members of all subgroups receive interventions. e) When false positives are expensive relative to false negatives and there are no fairness concerns, evaluation metrics should favor model improvements in decreasing order with score."_
>
> ### Q1: Can you provide analysis based on multiple subgroups?
> This is a great question and a rich area for future work. We've added to our Future Work section:
> "_Firstly, our theoretical findings can be refined and generalized to... specifically take into account more than 2 subpopulations for more nuanced comparisons beyond what can be inferred through pairwise comparisons between subpopulations, where our results would naturally apply_".

---

### Official Review · Reviewer_PmDV · 2024-07-09

**Soundness:** 3
**Presentation:** 2
**Contribution:** 3
**Rating:** 7
**Confidence:** 3

**Summary:**

The paper challenges the claim that area under the precision-recall curve (AUPRC) is a better metric for model comparison to the area under the receiver operating characteristic (AUROC) when it comes to tasks with class imbalance. The paper offers three formal results, proving i) a characterization of the two metrics in terms of how they relate to FPR, ii) a description of how the two metrics optimize for the correction of mistakes and iii) in fairness-sensitive settings, AUPRC introduces biases in favor of the highest prevalence subpopulations wrt AUROC. These findings are supported by synthetic and real-world experiments corroborating the findings.
They conclude that the aforementioned claim is unfounded and AUROC is preferable in several settings. Moreover AUPRC is potentially harmful in fairness-sensitive scenarios due to its bias towards the correction of mistakes in the highest prevalence subpopulations. Finally, the paper contains a review on the literature generating the claim and some guidance on when using which of the two metrics.

**Strengths:**

1. The paper combines theoretical results with empirical investigations on both synthetic and real-world data.
2. The paper does important work in challenging a claim that is widespread in the prediction model literature.
3. The practical implications of the theoretical findings, especially when it comes to fairness, are thoroughly investigated. The authors also research the origin of the claim and provide guidance on when to use which metric.
4. The text overall reads well and makes clear points. There are however some things to fix urgently in the presentation (which is why the presentation score is low and the overall score cannot be higher).

**Weaknesses:**

1.     Section 4 has a fair amount of repetition wrt Figure 1. I suggest compressing one or the other.
2.     Figure 2 is corrupted and the results cannot be assessed. This needs to be fixed.
3. The explanation is line 129-133 is not clear, I suggest re-writing it.
4.     A minor point, but I would suggest the authors to not use both boldface and italic at the same time (e.g. in the introduction). It feels unnecessarily intense.
5. Another minor point, but I would discourage the authors from using self-praising expressions such as “our analyses are thorough and compelling”. Let the reader be the judge of that.
6. In the caption of Figure 1, repeating the definition of mistake is not really necessary I believe.
7. The theoretical results have some strong assumptions, e.g. perfect calibration.
 Typos:
line 290: significnat -> significant
line 310: These -> these

**Questions:**

1.     For the relaxation of perfect calibration in Theorem 3, have the authors tried to consider different models with ascending levels of calibration, to see to what extent the property expressed by Theorem 3 holds?

**Limitations:**

The limitations seem to be properly discussed.

---

> ### Author Rebuttal · Authors · 2024-08-07
>
> Thank you for your comprehensive review and valuable comments!
>
> ### W1: Can you reduce the repetition in Section 4 and Figure 1?
> Thank you for this suggestion! We've condensed Figure 1's caption and Section 4. However, we maintain some overlap to ensure clear presentation of our key takeaway: understanding when to prefer AUROC or AUPRC as optimization metrics. To demonstrate these changes within the space limitations of the "official rebuttal", we provide our updated Figure 1 caption in response to your concern specifically about its redundancy later in this rebuttal.
>
>
> ### W2: Why does Figure 2 appear corrupted?
> We do not see any issues with Figure 2 on OpenReview, using Google Chrome's built-in PDF viewer. Could you please provide more details about the issue and your PDF viewer/OS in a comment? We will follow up promptly to ensure all figures are properly rendered.
>
> ### W3: Can you clarify the explanation in lines 129-133?
> In essence, we show that if one group of samples has a much higher rate of the outcome than another (e.g., men are more likely than women to receive a correct diagnosis for a heart attack), optimizing for AUPRC will provably preferentially optimize to fix mistakes that affect the high prevalence group over the low prevalence group (e.g., the model will preferentially learn to identify heart attack symptoms for men at the expense of learning those for women). In settings where such a model is used to determine who should receive some limited resource (e.g., to be evaluated by a cardiac specialist after an ED visit), this preferential optimization procedure will translate into a disparity in resource allocation between the two groups, which in many cases may be undesirable.
>
> In the text we have reworded this key result:
> _"Essentially, Theorem 3 (proof provided in Appendix F) shows the following.
> Suppose we are training a model $f$ over a dataset with two subpopulations: Population $a=0$ and $a=1$. If the model $f$ is calibrated and the rate at which $y=1$ for population $a=0$ is sufficiently low relative to the rate at which $y=1$ for population $a=1$, then the mistake that, were it fixed, would maximally improve the AUPRC of $f$ will be a mistake purely in population $a=1$. This demonstrates that AUPRC provably favors higher prevalence subpopulations (those with a higher base rate at which $y=1$) under sufficiently severe prevalence imbalance between subpopulations."_
>
>
> ### W4: Can you avoid using boldface and italics simultaneously?
> Thank you for this suggestion. We've removed all such instances.
>
> ### W5: Can you remove self-praising expressions?
> Thank you for this apt suggestion! We have removed all such instances.
>
> ### W6: Can you reduce redundancy in Figure 1's caption?
> Great suggestion. We have removed the repeated definition of the mistake from the figure caption. The whole caption of Figure 1 (which has been further condensed) is shown below:
>
> _"a) Consider a model $f$ yielding continuous output scores for a binary classification task applied to a dataset consisting of two distinct subpopulations, $\mathcal{A} \in \{0, 1\}$. If we order samples in ascending order of output score, each misordered pair of samples (e.g., mistakes 1-4) represents an opportunity for model improvement. Theorem 3 shows that a model's AUROC will improve by the same amount no matter which mistake you fix, while the model's AUPRC will improve by an amount correlated with the score of the sample. b) When comparing models absent a specific deployment scenario, we have no reason to value improving one mistake over another, and model evaluation metrics should therefore improve equally regardless of which mistake is corrected. c) When false negatives have a high cost relative to false positives, evaluation metrics should favor mistakes that have *lower scores*, regardless of any class imbalance. d) When limited resources will be distributed among a population according to model score, *in a manner that requires certain subpopulations to all be offered commensurate possible benefit from the intervention for ethical reasons*, evaluation metrics should prioritize the importance of within-group, high-score mistakes such that the highest risk members of all subgroups receive interventions. e) When false positives are expensive relative to false negatives and there are no fairness concerns, evaluation metrics should favor model improvements in decreasing order with score."_
>
> ### W7: How do you address the strong assumptions in theoretical results, e.g., perfect calibration?
> This is true; We acknowledge this limitation and have extended our discussion in Section 6: _"In addition, one of the largest limitations of Theorem 3 is its restrictive assumptions, in particular the requirement of perfect calibration. A ripe area of future work is thus to investigate how we can soften our analyses for models with imperfect calibration or to determine whether or not our results imply anything about the viability or safety of post-hoc calibration of models optimized either through AUPRC or AUROC"_
>
> ### W8: Can you fix the typos on lines 290 and 310?
> Thank you for catching these! We have corrected both.
>
> ### Q1: Have you considered models with ascending levels of calibration?
> This is a great area for future work. We believe it's possible but requires careful specification of model miscalibration. For example, bounding a model's calibration error over any region of the output score space could help bound the extent to which prevalence gaps correspond to different prediction rates, translating into bounds on AUPRC's preference for fixing high-prevalence group mistakes. We've highlighted this in Section 6 with strengthened language.

---

> > ### Comment · Reviewer_PmDV · 2024-08-09
> > **Response to rebuttal**
> >
> > I thank the authors for the rebuttal, which addresses my points. Concerning Figure 2: this is visualized as corrupted when the file is opened on Safari; I see this was not a problem for the other reviewers but I am nonetheless surprised since this never happened with other papers on OpenReview. I recommend the authors double-check the formatting of the image in their iteration on the paper.

---

> > > ### Author Response · Authors · 2024-08-09
> > > **Image Corruption Reproduced; we will address**
> > >
> > > Thank you for your impressively quick response, and for providing the additional details on the image corruption in Figure 2. We have successfully replicated this issue by viewing it in Safari, and can confirm that this is very different and clearly corrupted in comparison to how we see the image in, for example, Chrome. Now that we can reproduce this issue, we will debug and correct it promptly. Thank you again for providing these additional details!

---

> > > > ### Comment · Reviewer_PmDV · 2024-08-09
> > > > **Image corruption**
> > > >
> > > > Thanks for looking into this. I would appreciate if you could upload the image to your general rebuttal in a format that I can visualize in Safari. This way I am sure I have seen the paper in its entirety.

---

> > > > > ### Author Response · Authors · 2024-08-09
> > > > > **Link to image file sent to AC**
> > > > >
> > > > > Absolutely; we're happy to send the figure in a different format. Per the [NeurIPS 2024 FAQ for authors](https://neurips.cc/Conferences/2024/PaperInformation/NeurIPS-FAQ#:~:text=the%20author%20rebuttal%3F-,No.,all%20linked%20files%20are%20anonymized), we're told not to send links in any part of the response, and unfortunately we do not have the ability to edit the official rebuttal anymore, only post comments, so we can't upload a PDF with the image in a different format now. But, we've sent a message to the AC with an anonymized link to the image in a png format directly that we can confirm we see in Safari in full; I believe the AC should be able to share that link with you (this is how we are instructed to share links in the instructions). Thank you again for helping us debug this and for your understanding!

---

### Official Review · Reviewer_2iwp · 2024-07-10

**Soundness:** 3
**Presentation:** 4
**Contribution:** 3
**Rating:** 8
**Confidence:** 4

**Summary:**

The paper proves that AUPRC weights mistakes in higher score ranges higher, while AUROC weights all mistakes uniformly. This property of AUPRC can be underable in many real-life settings. It goes against the widespread belief that AUPRC is somehow “better” than AUROC in low-prevalence domains, which is a common belief both in academia and industry.

The paper also proves a theorem that states that AUPRC is be discriminatory against low-scoring subpopulations, and it backs up this theoretical finding by a series of experiments that confirm that these discriminatory effects occur in practice when selecting models based on AUPRC

**Strengths:**

It has long been well-established in the literature that the AUROC is prevalence-invariant (or as they put it in concept drift-focused ML subcommunities: it is invariant to prior probability shift), while the AUPRC is not [1,2,3,4]. In the light of this, the argument that AUPRC would somehow be “better” than AUROC in low-prevalence domains has always appeared paradoxical to me (how could a prevalence-invariant metric be unsuitable in low-prevalent domains?).

This paper does an excellent job in theoretical analysis of AUPRC and AUROC, an engaging read, and is clear on the practical implications. Figure 1 provides a compelling overview of the main results and manages to convey the implications clearly in one glance. I really enjoyed this read and consider this to be an important paper.

**Weaknesses:**

A minor critique goes to the experimental section. In both the synthetic and the real-world evaluation the confidence intervals appear rather wide. While the Figures 2 and 3 do appear to support the message that the authors claim that it does, it could be more convincing. For example,
Is it really true that the AUROC for the low-prevalent sub-group decreases in Figure 2d, or is this noise?
The AUROC of the high-prevalence group in Figure 2b appears to trend up while increasing steps while the one of the low-prevalence group appears to stay flat, as the theory would suggest. However, the confidence intervals do still overlap everywhere.
Most of the CIs for the correlations in Figure 3 include 0.0.

This appears to simply be a matter of sample size: 20 randomly sampled datasets in the synthetic evaluation and relatively small tabular datasets for the real-life data evaluations.
It appears that Figure 2 may become more convincing simply by simulating more randomly sampled datasets (reducing the standard error of the mean), while for Figure 3 we may obtain a more convincing plot by simply including a larger real-life dataset.

That said, I consider this to be a minor point. The main contributions of the paper are theoretical, and despite my critique, the current figures do support the theory. Just not as convincingly as could have been, and it appears easily solvable.

**Questions:**

- Line 199/200: “We also evaluate the test set AUROC gap and AUPRC gap between groups”. I believe this may be a mistake, as Figures 3, 7, and 8 only appear to report AUROC gaps and not AUPRC gaps. I believe it is the right choice to refrain from reporting AUPRC gaps: the fact that AUPRC is known to be prevalence-dependent would imply that prevalence-gaps by themselves may already explain AUPRC-gaps, even without existence of underlying fairness issues. The experimental setup based on AUROC gaps appears correct.
- Line 154: "such that $AUROC(D1) \approx AUROC(D2) \approx AUROC(D1\cup D2) = 0.85$. What is the algorithmic procedure that is used to obtain scores at the target AUROC, and how precisely does this achieve the target AUROC?
- Line 163: "Next, we profile an optimization procedure that randomly permutes all the (sorted) model scores up to 3 positions." This description seems imprecise, what is the precise reordering procedure that is applied?

**Limitations:**

The authors adequately addressed the limitations and potential negative societal impact of their work.

---

> ### Author Rebuttal · Authors · 2024-08-07
>
> Thank you for your time, expertise, recognition of the impact of our work, and helpful suggestions. Below, we address each of your questions or concerns individually.
>
> ### W1: Why is the variance so high in the experiments?
> Two factors contribute:
> 1. We report confidence intervals spanning the 5th to 95th percentiles, not standard errors of the mean; these confidence intervals will, therefore, remain wide no matter how many samples we run and not shrink towards the mean as some other measures of variance do.
> 2. Our procedure samples from an extremely wide space of possible models, resulting in high true variance by design. This is important so we can make sufficiently general conclusions about the validity of our theory.
>
> We've added the following to Figure 2's caption to help clarify: "_Synthetic experiment per-group AUROC, showing a confidence interval spanning the 5th to 95th percentile of results observed across all seeds_" to help clarify this.
>
> For the real-world data, these experiments are actually somewhat expensive to run, as we sample a very large number of models, seeds, and hyperparameter options in order to sufficiently assess the correlations of the fairness disparities under AUROC and AUPRC, so using larger datasets does pose an additional challenge there as well. Relatedly, the datasets we have chosen are some of the more common fairness datasets, so without branching into much more intensive modeling domains, we have limited options for additional datasets that are widely used.
>
> That said, your point is still well taken and we will explore other ways to present these results in our revision to better demonstrate the consistency and statistical reliability of aggregate results across many samples, while still communicating the raw variance.
>
> ### Q1: There is a mistake on line 199/200.
> Thank you for catching this typo! We have corrected this sentence to be _"We also evaluate the test set AUROC gap between groups, where gaps are defined as the value of the metric for the higher prevalence group minus the value for the lower prevalence group."_
>
> ### Q2: What Algorithmic Procedure Obtains Scores at the Target AUROC?
> We outline the procedure in Appendix G.1. Briefly, we randomly sample scores for positively labeled samples, then draw scores for negatively labeled samples under a distribution that ensures, in expectation, the count of positive samples with scores above or below any given negative sample will be precisely the target AUROC. While exact for single-group AUROC, it is not necessarily exact for multi-group AUROC constraints, though it works well in practice.
>
> To improve clarity, we have added the following text to our paper:
>   1. On line 154, the sentence now reads _"... such that $\text{AUROC}(\mathcal D_1) \approx \text{AUROC}(\mathcal D_2) \approx \text{AUROC}(\mathcal D_1 \cup \mathcal D_2) = 0.85$ (See Appendix G.1 for technical details; ...)."_
>   2. In Appendix G.1, we have added a new paragraph at the end which states _"The procedure outlined above guarantees that, in expectation, the AUROC of the generated set of scores and labels will be precisely the target AUROC. However, if you apply this procedure independently across different sample subpopulations, this guarantee can only be applied on each subpopulation individually, and not necessarily on the overall population due to the unspecified xAUC term. However, in practice, for the experiments we ran here, that impact neither meaningfully impacts our experiments nor were the joint AUROCs sufficiently different from the target AUROC to warrant a more complex methodology."_
>
> ### Q3: What is the precise reordering procedure applied?
> This procedure is described in detail in Appendix G.3 of the submitted Manuscript, subsection "M3. Sequentially Permuting Nearby Scores." In short, the reordering procedure selects a random permutation of scores, realized as a permutation matrix, such that that permutation matrix has no entries more than 3 positions off the diagonal (thus ensuring that samples cannot change position by more than 3 places). We have amended line 163 to reference this, stating _"... (sorted) model scores up to 3 positions (See Appendix G.3 for details)."_ The implementation of this procedure is released along with the rest of our experimental code at the link included in the manuscript. If requested, we are happy to add more details about the procedure we use to the text or the comments here.

---

> > ### Comment · Reviewer_2iwp · 2024-08-12
> >
> > I thank the authors for the clarifications. My score, which was already very positive, remains unchanged.

---

> > > ### Author Response · Authors · 2024-08-13
> > >
> > > Thank you for your response, for your excellent review, and for going through our clarifications!

---

### Official Review · Reviewer_XQD9 · 2024-07-12

**Soundness:** 3
**Presentation:** 2
**Contribution:** 3
**Rating:** 6
**Confidence:** 3

**Summary:**

The paper considers a common in literature claim that AUPRC is “better” to use than AUROC for class imbalance datasets and attempts to prove it wrong based on theoretical results, empirical observations, and real-world experiments. The major focus of the paper is on the fairness gap that AUPRC exhibits. The authors provide guidance on when each of the metrics should be used and advocate for careful metric selection.

**Strengths:**

1. The paper provides a thorough review of the literature, aiming to get to the root cause of Claim 1, and advocates for more responsible practices in choosing evaluation metrics.
2. The authors attempt to examine the problem from different perspectives, including theoretical analysis, synthetic, and empirical experiments.
3. The paper includes extensive details on notation, definitions, experimental details, and figures.

**Weaknesses:**

1. It is unusual to see in the synthetic example in Section 3 that AUPRC is being optimized for the classification setting (as it is more common to use AUPRC as an evaluation metric). Could you please elaborate on when it can potentially be useful?
2. For the synthetic dataset example, since everything is controllable, including the number of mistakes, does it make sense to learn boosted trees or any other model of the authors' choice for different levels of mistakes and report AUROC/AUPRC instead of the proposed optimized procedure?
3. Since AUPRC focuses on the positive class and does not use true negatives, it is "expected" by definition of this metric to increase existing fairness gaps in the positive (minority) class. Could you please elaborate on what Theorem 3 adds beyond the definition?
4. The incorrectly ranked adjacent pair mistake model seems to favor AUROC over AUPRC. By definition, AUROC measures the ability of the model to distinguish between positives and negatives across all possible thresholds, so fixing this mistake improves AUROC uniformly. By definition of AUPRC, correcting such a mistake can have a significant or minor impact depending on the threshold.
5. While p-values and checking for for statistical significance is important, in addition for the experimental results on real-world datasets, could you please provide actual values of AUROC and AUPRC (mean and variance), for example in a bar plot, similar to what is typically reported in fairness literature?
6. The broader idea of the paper—to advocate for the careful selection of metrics—is highly welcomed. AUROC and AUPRC are different metrics with different goals, just as ROC and PR curves are. They can provide different insights when evaluating data with class imbalance. Wouldn’t it make sense to encourage readers to explore both AUROC and AUPRC, as well as ROC and PR curves, under class imbalance, given the authors' findings and reference [83] instead of proposing to use one metric as in Section 4?

Minor:
Some figure legends seem to be cut off in the appendix figures.

**Questions:**

1. What causes the variance to be so high in Figure 2?
2. Could you please elaborate more on what 'Prevalence (Higher)' and 'Prevalence (Lower)' measure in Table 1? If class imbalance, shouldn’t they sum up to 1?

**Limitations:**

The authors discuss the limitations of the paper.

---

> ### Author Rebuttal · Authors · 2024-08-07
>
> Thank you for your insightful and comprehensive review! Note that, for space reasons, we respond to your two questions in a "comment" rather than here in our official "rebuttal."
>
> ### W1: Why optimize for AUPRC?
> AUPRC is implicitly used as an optimization metric in cases where it is the target for hyperparameter tuning or model selection (as we explore in our real-world experiments). It also can be an explicit optimization target via specialized algorithms (e.g., [here](https://proceedings.neurips.cc/paper_files/paper/2022/hash/b5dc49f44db2fadc5c4d717c57f4a424-Abstract-Conference.html), [here]( https://proceedings.neurips.cc/paper_files/paper/2021/file/0dd1bc593a91620daecf7723d2235624-Paper.pdf)).
>
> Our work shows that optimizing for AUPRC favors subpopulations with higher outcome base rates. We've added clarification to Section 3: "_Simulating optimizing by these metrics allows us to explicitly assess how the use of either AUPRC or AUROC as an evaluation metric in model selection processes such as hyperparameter tuning can translate into model-induced inequities in dangerous ways._"
>
> ### W2: Why not use boosted trees for the synthetic example?
> While in our real-world experiments, we use boosted trees for exactly the reasons you highlight, in our synthetic experiments we need to use more controlled optimization procedures so that we can precisely attribute the results to the choice of optimizing via AUROC or AUPRC in isolation. In particular, using a classical model risks confounding the impact of AUROC vs. AUPRC with factors such as the ease or difficulty of optimizing towards the target for various subpopulations, optimization algorithms, or output score regions.
>
> To make this clearer, we have added to Section 3.1: "_In this section, we use a carefully constructed synthetic optimization procedure to demonstrate that, when all other factors are equal, optimizing by or performing model selection on the basis of AUPRC vs. AUROC risks exacerbating algorithmic disparities in the manner predicted by Theorem 3. For analyses under more realistic conditions with more standard models, see our real-world experiments in Section 3.2._"
>
> ### W3: What does Theorem 3 add beyond the definition of AUPRC?
> Our work's theoretical analyses provide several key insights:
>   1. Theorem 1 demonstrates that AUPRC & AUROC can be expressed as very similar linear functions of the expectation of the model's FPR over positive class scores. This reveals that the key difference between AUROC & AUPRC is not precisely that AUPRC focuses more on the positive class or doesn't care about true negatives, but rather that AUPRC weights model errors more heavily when they occur in high-score output regions vs. low-score regions, whereas AUROC weights all errors equally.
>   2. Theorem 3 formalizes the intuition that you rightly identify: that given AUPRC's focus on the high-score region, subpopulations with a higher prevalence of the label will be preferentially optimized by AUPRC. To the best of our knowledge, this has not been previously proven formally.
>   3. Critically, these theoretical formulations demonstrate both the widespread misconception in the ML community that AUPRC is generally superior for skewed data (Section 5) and formally prove why this misconception is dangerous to model fairness. In particular, despite the intuitive recognition that some top experts like you may have about AUPRC's fairness risks, many authors have used AUPRC as a selection and principal evaluation metric in fairness-centric settings, as noted in a variety of the papers in Section 5.
>
> ### W4: Doesn't the mistake model favor AUROC over AUPRC?
> We agree that correcting adjacent pair mistakes can have varying impacts on AUPRC depending on the threshold. This behavior--and critically, its consequences on model fairness and appropriate use cases--is precisely what we want to highlight to readers in our work. In particular, this model makes it clear that in high-score regions and retrieval contexts without fairness constraints, AUPRC is appropriate. For optimization problems that heavily depend on lower-score regions, AUPRC is less appropriate. This holds _regardless of class imbalance_, despite the widespread belief in the community that AUPRC should be generally preferred in cases of class imbalance.
>
>
> ### W5: Can you provide actual AUROC and AUPRC values?
> Yes, we will make these results available in our revision. However, we have two notes about these raw values:
>   1. Our experimental analyses are focused on the impact of performing model selection or other optimization by AUROC vs. AUPRC under varying levels of prevalence imbalance. This question is therefore inherently not about the precise AUROC and AUPRC disparities in a single model run, but rather about how optimizing by AUROC vs. by AUPRC impacts fairness gaps in aggregate across many hyperparameter settings and prevalence disparities.
>   2. Accordingly, the number of "actual AUROC and AUPRC values" we have is quite large. For our experiments, we train a very large number of models (500+) over different hyperparameters and across many seeds so that we can assess the impact of the evaluation choice with sufficient statistical power. Nevertheless, we will absolutely make all these results available in our revised manuscript.
>
> ### W6: Shouldn't readers explore a variety of metrics?
> You're right; to clarify this, we have added the following to the top of Section 4: "_Note that while we provide guidance below on situations in which AUROC vs. AUPRC is more or less favorable, this is not to suggest that authors should not report both metrics, or even larger sets of metrics or more nuanced analyses such as ROC or PR curves; rather this section is intended to offer guidance on what metrics should be seen as more or less appropriate for use in things like model selection, hyperparameter tuning, or being highlighted as the 'critical' metric in a given problem scenario._"

---

> ### Author Response · Authors · 2024-08-07
> **Additional comments (in addition to the official rebuttal) for reviewer XQD9's questions**
>
> Thank you for your many insightful and helpful comments to improve our work! Note that, for space reasons, while we have responded to all your identified weaknesses individually in our official "rebuttal" (which may or may not be released to reviewers at the time this comment is visible, depending on openreview settings), in this "comment" we respond to your questions instead. Please see the rebuttal as well for our full response to your excellent review.
>
> ### Q1: What causes the high variance in Figure 2?
> Two factors contribute:
> 1. We report confidence intervals spanning the 5th to 95th percentiles, not standard errors of the mean; these confidence intervals will therefore remain wide no matter how many samples we run, and not shrink towards the mean as other measures of variance do.
> 2. Our procedure samples from an extremely wide space of possible "models" resulting in high true variance by design. This is important so we can make sufficiently general conclusions about the validity of our theory.
>
> We've added to Figure 2's caption: "_Synthetic experiment per-group AUROC, showing a confidence interval spanning the 5th to 95th percentile of results observed across all seeds_" to help clarify this.
>
> ### Q2: What do 'Prevalence (Higher)' and 'Prevalence (Lower)' mean?
> These refer to the fraction of samples with label 1 in the subpopulations with the highest and lowest prevalence, respectively. They don't sum to 1 as they're for different subpopulations. E.g., if we are predicting the likelihood a patient has had a heart attack based on their symptoms for a population containing both men and women, both subgroups will have different rates of heart attacks, and those rates will not sum to one.
>
> To clarify, we've added to Table 1's caption:
> "_Here, ``Prevalence (Higher)'' refers to the rate at which the prediction label $y=1$ for the subpopulation with a higher such rate, and ``Prevalence (Lower)'' refers to the same rate but over the subpopulation of the dataset with a lower rate of $y=1$._"

---

> > ### Comment · Reviewer_XQD9 · 2024-08-12
> >
> > Thank you to the authors for the rebuttal. After carefully reading the reply, the paper’s contributions are clearer to me, so I will increase my score and recommend acceptance.
> >
> > I encourage the authors to review the paper carefully to improve its clarity.
> >
> > > Accordingly, the number of "actual AUROC and AUPRC values" we have is quite large.
> >
> > For Figure 3, you can report the mean and variance over different text/train splits for cross-validated parameters. I believe this should strengthen the paper's results.
> >
> > > We agree that correcting adjacent pair mistakes can have varying impacts on AUPRC depending on the threshold.
> >
> > Adding this clarrification to the paper will help readers better understand the mistake model and its role.

---

> > > ### Author Response · Authors · 2024-08-12
> > >
> > > Thank you for going through our rebuttal so clearly and for raising your score! Per your suggestion, we will definitely add a summary of the raw results as well to our work, and will add clarifying text to the manuscript to help readers better understand the mistake model and why it shows such different performance under AUROC vs. AUPRC. Thank you for both of these excellent suggestions!

---

> ### Author Response · Authors · 2024-08-12
> **Request for any additional feedback or concerns given our rebuttal**
>
> Thank you again for your time and valuable feedback! As the rebuttal period comes to a close, we were wondering if our response has adequately addressed your concerns. If there are any remaining questions or comments, we would be happy to discuss!

---

### Author Rebuttal · Authors · 2024-08-07

We sincerely thank all reviewers for their insightful, comprehensive, and constructive feedback! We're particularly encouraged by the widespread recognition of our work's novelty, impact, and theoretical contributions, with reviewers noting things like
  * "I really enjoyed this read and consider this to be an important paper." _(Reviewer 2iwp)_
  * "This paper does important work in challenging a claim that is widespread in the prediction model literature." _(Reviewer PmDV)_
  * "The proposed ideas are novel and innovative. Theoretically, the authors explore the relationship between AUROC and AUPRC, revealing their key differences" _(Reviewer pyKs)_

In addition, we found the various points of feedback extremely helpful and have, based on your comments, made the following key improvements:
  1. **Systemic Presentation Improvements**: While we were glad to see some aspects of our presentations highlighted positively, reviewers were generally aligned on a need for improvements to our presentation. Accordingly, we have
     - Extensively condensed the caption of Figure 1 and reduced redundancy between Figure 1 and Section 4, while maintaining their clarity and informative value _(as suggested by Reviewers PmDV and pyKS)_.
     - Clarified our usage of "prevalence" across multiple subpopulations in the context of Theorem 3, significantly enhancing the clarity of one of our key theoretical results _(as suggested by Reviewers 2iwp, PmDV, and pyKS)_.
     - Addressed any missing experimental details, typos, poor word/formatting choice, missing citations or references to appendix sections, and added raw experimental results _(as suggested by Reviewers XQD9, 2iwp, PmDV, and pyKS)_.

  2. **Clarification of the Novelty and Import of our Findings**: While reviewers 2iwp, PmDV, and pyKS all noted specifically the novelty and importance of our findings, we also made significant improvements to these areas as well in line with the many suggestions, including:
     - Expanding our commentary on how our findings extend upon the raw definition of AUPRC and on why it matters that AUROC and AUPRC behave differently under the "mistake correction" model of optimization _(as suggested by Reviewer XQD9)_.
     - Clarified the notion of "AUPRC optimization" in our work and our work's relationship to past works on AUPRC optimization _(As suggested by Reviewers XQD9 and pyKS)_.

We're confident these changes address the main concerns raised and strengthen the paper significantly. Each reviewer's concern is addressed in more detail below. Please do not hesitate to respond to our rebuttals or leave additional comments if you have more feedback or questions or want to see further revisions to address your concerns. Thank you to all reviewers again.

---

### Decision · Program_Chairs · 2024-09-25

**Decision:**

Accept (poster)

**Comment:**

This paper provides a comprehensive theoretical and empirical rebuttal of the folklore assertion that area under the precision-recall curve (AUPRC) is a superior metric for model comparison as compared to the more standard area under the receiving operator characteristic (AUROC) in situations where there is class imbalance. Ultimately, all reviewers were in support of acceptance of this paper and found the contributions to be innovative and valuable to the ML community. I concur with this recommendation.

Please address all suggestions made by the reviewers to improve readability and figure formatting in the camera-ready version.